# Machinability and tribological optimization of origami-inspired Almond Shell–PMMA via RSM, ML, and TOPSIS

Biplab Bhattacharjee[1], S. Sivalingam[2], Raman Kumar[3,4], Jasgurpreet Singh Chohan[5], Sandeep V.[6], Ripendeep Singh[7], Anupama Routray[8], Jibitesh Kumar Panda[9]*

**1** Department of Mechanical Engineering, Faculty of Engineering and Technology, SRM Institute of Science and Technology, Ramapuram, Chennai, Tamil Nadu, India, **2** Department of Mechanical Engineering, J. J. College of Engineering and Technology, Tiruchirappalli, Tamilnadu, India, **3** University School of Mechanical Engineering, Rayat Bahra University, Kharar, Punjab, India, **4** Faculty of Engineering, Sohar University, Sohar, Oman, **5** Marwadi University Research Center, Department of Mechanical Engineering, Faculty of Engineering & Technology, Marwadi University, Rajkot, Gujarat, India, **6** Department of Mechanical Engineering, School of Engineering and Technology, JAIN (Deemed to be University), Bangalore, Karnataka, India, **7** Department of Mechanical Engineering, University Institute of Engineering and Technology, Chandigarh University, Mohali, India, **8** Sharda School of Engineering and Sciences, Sharda University, Greater Noida, Uttar Pradesh, India, **9** Manipal Institute of Technology, Manipal Academy of Higher Education, Manipal, Karnataka, India

* jibitesh.panda@manipal.edu

## Abstract

An integrated approach combining Response Surface Methodology (RSM), Machine Learning (ML-SVM) and TOPSIS optimization method is applied in this study to analyse the tribological behaviour of 3D printed patterns of almond shell-PMMA (polymethyl methacrylate) origami inspired composites and machinability. The process of taking advantage of fold-based geometrical patterns (such as Miura-ori or triangular tessellation) to enhance load distribution and energy absorption in 3D printed specimen is called origami-inspired. These patterns promote a certain degree of structural rigidity and cause weakened materials to deform under applied stress in a controlled manner in general to improve the mechanical strength and wear-resistance. Besides tribological performance factors such as wear rate and friction coefficient, the factors to be evaluated on the machinability properties of cutting force, surface roughness and material removal rate include spindle speed (3000−9000 rpm), feed rate (0.05–0.15 mm/rev), and depth of cut (0.2–0.6 mm). Although the machine learning algorithms were able to make predictive models concerning wear performance and machinability, RSM was addressed to plan the experiments and conclusion of the parameters interaction. TOPSIS method identified the parameters combination that will serve the best by balancing between tribological efficiency and machinability. The novelty aspect of the current work is the inclusion of agricultural waste (10 percent almond shells) to the polymer matrices and and the use of a hybrid optimization strategy on the ways to optimize its functional properties with respect to being used

**Data availability statement:** All raw data required to replicate the findings of this study, including the numerical values behind all figures, tables, RSM models, ML datasets, and TOPSIS matrices, are fully included within the manuscript and its Supporting information files. No additional external dataset was generated or used.

**Funding:** The author(s) received no specific funding for this work.

**Competing interests:** The authors have declared that no competing interests exist.

in wear and machining applications. Notable findings indicate that the most influential variable affecting machinability and tribological results is the feed rate; in the best case scenario there is achievement of surface roughness of 1.2 10−15 m and wear rate aside at 1.5 in 10−4 mm 3/Nm. The proposed model was able to give a workable and industrially friendly composite method with great efficiency and sustainability in terms of optimization performance and predictability ($R^2 > 0.95$).

## 1. Introduction

The increasing demand for sustainable materials has driven significant research into bio-based reinforcements for polymer composites. Among these, agricultural waste materials have gained attention due to their environmental benefits, cost-effectiveness, and potential for enhancing the mechanical and tribological properties of polymer matrices [1]. Almond shells, a byproduct of the almond industry, are rich in lignocellulosic content and possess excellent mechanical properties, making them a promising reinforcement for polymer composites. Their integration into Poly Methyl Methacrylate (PMMA) matrices offers a viable approach to improving machinability and tribological behaviors, particularly for applications requiring wear-resistant coatings and structural integrity [2,3]. Machinability, defined as the ease of material removal while maintaining surface integrity, dimensional accuracy, and tool longevity, is a crucial factor in material processing. In polymer matrix composites (PMCs), machining characteristics are influenced by factors such as fiber content, matrix composition, tool geometry, and cutting parameters (spindle speed, feed rate, and depth of cut) [4–6]. Studies [4,7–9] have demonstrated that natural fiber-reinforced polymers (NFRPs) exhibit superior machinability compared to synthetic fiber-reinforced composites due to their lower hardness and enhanced thermal stability. However, the introduction of bio-based fillers, such as almond shells, alters these properties, necessitating systematic optimization techniques to achieve desirable machining outcomes.

Response Surface Methodology (RSM) has been extensively utilized in machining studies to optimize process parameters while minimizing experimental costs. RSM-based models provide valuable insights into the interactions between machining parameters and performance indicators such as material removal rate (MRR), cutting force, and surface roughness (Ra). Previous research has highlighted that among machining parameters, feed rate plays a dominant role in determining surface roughness and tool wear, while spindle speed influences heat generation and chip formation [10–12]. Tribological performance, encompassing wear resistance and friction coefficient, is a critical aspect of polymer composites, particularly in applications involving sliding or abrasive contact. The use of bio-based fillers such as nutshell reinforcements has been shown to enhance wear resistance due to their inherent lubricating properties and fiber-rich structures [13]. Studies on nutshell-based composites, including walnut and coconut shells, have demonstrated their ability to reduce wear rates and improve frictional stability under dry sliding conditions.

Given the hardness and cellulose-rich composition of almond shells, their integration into PMMA is expected to provide enhanced tribological performance. However, factors such as filler dispersion, interfacial bonding, and operational conditions significantly influence the wear behavior of these composites, necessitating empirical and computational analyses [14–17].

Machine Learning (ML) has emerged as a powerful tool in materials engineering for predicting and optimizing machining and tribological properties. Advanced ML algorithms, including Artificial Neural Networks (ANNs), Support Vector Machines (SVMs), and Random Forests, have demonstrated superior accuracy in capturing nonlinear relationships between process parameters and material responses [18–21]. Studies have shown that ML models outperform traditional regression approaches in predicting surface roughness, wear rate, and cutting forces, enabling more precise decision-making in material processing. In recent years, ML-based hybrid models have been employed to enhance the predictive accuracy of machining and tribological behaviours. By training on experimental datasets, ML algorithms can effectively model complex interactions and provide real-time insights into material performance. This study integrates ML techniques to develop predictive models for machinability and wear performance of almond shell-PMMA composites, offering a data-driven approach for optimizing composite properties [22,23].

The Technique for Order of Preference by Similarity to Ideal Solution (TOPSIS) is a widely used Multi-Criteria Decision-Making (MCDM) technique that ranks alternative solutions based on their closeness to an ideal solution [24–26]. In machining and tribological studies, conflicting objectives such as minimizing surface roughness while maximizing wear resistance require a balanced optimization approach. TOPSIS has been successfully applied to identify optimal process parameters that satisfy multiple performance criteria, providing a structured methodology for selecting the best machining conditions [27,27]. By integrating RSM, ML, and TOPSIS, this study develops a comprehensive optimization framework for enhancing the machinability and tribological performance of almond shell-PMMA composites. The novelty of this work lies in the synergistic combination of experimental design, predictive modelling, and decision-making techniques to establish an efficient, sustainable, and high-performance composite solution for industrial applications [28,29].

The reviewed studies span advances in biomaterials, coatings, additive manufacturing, and smart material design. Cao et al. [30] developed robust lubricative core–shell nanofibers to suppress friction-induced adhesion, improving tendon repair quality. Liu et al. [31] investigated the tribocorrosion resistance of Inconel 625 composite coatings reinforced with $Al_2O_3$ using plasma-enhanced arc spraying, revealing enhanced durability. Huang et al. [32] and Dong et al. [33] focused on wire-based friction stir technologies, with the former demonstrating effective additive remanufacturing of 2219 Al alloy components, and the latter achieving equal-strength aluminium alloy welding even with assembly gaps. Cen et al. [34] combined Nano indentation, machine learning, and microstructure-informed modeling to construct global yield surfaces of polymethacrylimide foam, offering a predictive framework for mechanical behaviour. Zhang et al. [35] advanced robotic wire-based friction stir additive manufacturing, highlighting automation and scalability. Finally, Zha and Zhang [36] introduced metamaterials with reversible negative compressibility inspired by Braess's Paradox, expanding the frontier of smart structural design. Together, these works highlight innovations in tribology, coating durability, additive manufacturing, and multifunctional material development for advanced engineering and biomedical applications.

The novelty of this study lies in three distinct aspects: (i) the valorization of agricultural waste by incorporating almond shell powder into PMMA for sustainable composite development, (ii) the design of origami-inspired 3D printed patterns that enhance machinability and tribological performance, and (iii) the integration of a hybrid optimization framework—combining RSM for experimental design, Machine Learning (SVM) for predictive modelling, and TOPSIS for multi-objective decision-making. This unique combination of sustainable reinforcement, innovative structural design, and advanced hybrid optimization has not been reported in prior studies, thereby establishing the originality of the present work.

The utilization of almond shell waste as a reinforcement in PMMA composites presents a promising avenue for sustainable material development. By leveraging RSM for experimental optimization, Support Vector Machines (SVMs) for predictive modelling, and TOPSIS for decision-making, this research provides a robust framework for enhancing machinability

and tribological efficiency. The findings of this study contribute to the broader efforts in waste valorization and computational modelling in material science, paving the way for future advancements in eco-friendly polymer composites.

## 2. Material and methodology

### 2.1 Material selection and preparation

The primary materials used in this study include Poly Methyl Methacrylate (PMMA) as the matrix material (as illustrated in Fig 1(a) and 1(b)) and almond shell powder as the reinforcement (as depicted in Fig 1(c) and 1(d)). The almond shells were sourced from agricultural waste, cleaned thoroughly to remove any impurities, and then dried at 80°C for 24 hours to eliminate residual moisture. Once dried, the almond shells were ground into fine powder using a high-energy ball milling process. This method ensures a uniform particle size distribution, crucial for maintaining homogeneity in the composite structure. The ground powder was then sieved using a 100-micron mesh to obtain a consistent particle size range between 50 and 100 microns, ensuring effective dispersion in the PMMA matrix.

To improve interfacial bonding between the almond shell particles and PMMA, surface treatment was carried out using a silane coupling agent. The almond shell powder was immersed in a 2% silane solution and subjected to ultrasonication for 30 minutes to enhance the chemical interaction between the filler and polymer matrix. Following treatment, the powder

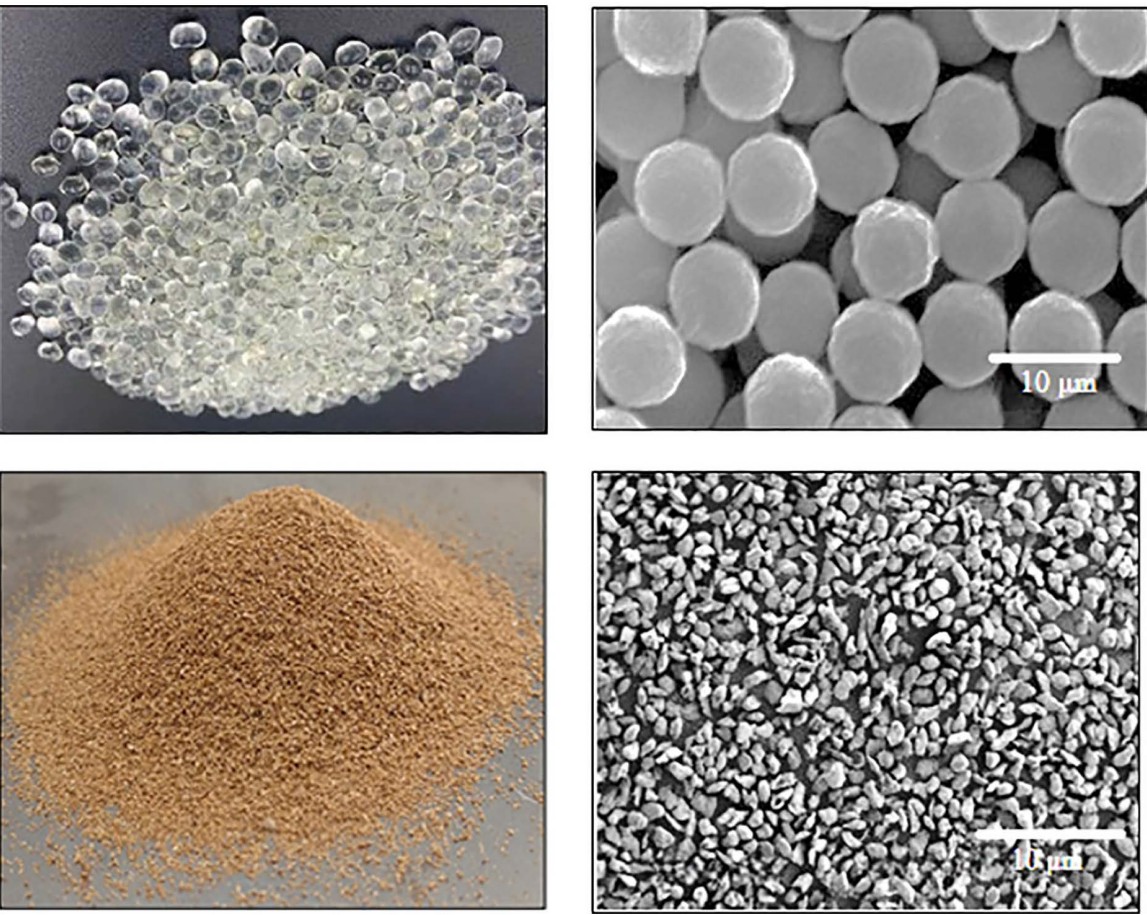

**Fig 1. Actual and SEM images of (a,b) PMMA and (c,d) Almond shell powder.** Scale bar = 10 μm for SEM images.

was dried again to remove any residual solvent before composite fabrication. The PMMA matrix was selected due to its excellent mechanical properties, lightweight nature, and ease of processing in 3D printing applications. To ensure compatibility with the Fused Deposition Modelling (FDM) process, the PMMA was first dissolved in a suitable solvent and mixed with the almond shell powder to form a uniform composite mixture. This mixture was then processed into filament form using a twin-screw extruder, maintaining a consistent filament diameter of 1.75 mm for smooth printing performance. This study was conducted entirely within laboratory facilities, and no field site access or sampling was involved. Hence, no external permits were required.

## 2.2 Fabrication of Almond Shell-PMMA composites

The composite samples were prepared using Fused Deposition Modelling (FDM) 3D printer as depicted in Fig 2(a) and 2(b). The fabrication process involved several key steps:

- **Filament Preparation:** The almond shell powder was mixed with PMMA resin and extruded into filament form using a twin-screw extruder. The extruded filaments were cooled and wound onto spools for use in 3D printing.

- **3D Printer Setup:** A commercial FDM 3D printer was employed to print the test specimens. The printer settings, including nozzle temperature (230–250°C), bed temperature (90–110°C), layer height (0.1–0.3 mm), and infill density (50–100%), were optimized based on preliminary trials.

- **Printing Process:** The prepared filament was fed into the FDM printer, and specimens were printed layer by layer using a predefined pattern to achieve the required dimensions and mechanical properties.

- **Post-Processing:** After printing, the specimens underwent heat treatment in a hot air oven at 80 °C for 2 hours to relieve residual stresses and enhance interfacial bonding between the matrix and reinforcement. Surface finishing techniques such as sanding and polishing were applied to ensure uniformity and accuracy in specimen dimensions.

Fig 3(a) presents the SEM image of pure PMMA, revealing a relatively smooth surface with minor irregularities. In contrast, Fig 3(b) shows the SEM image of PMMA reinforced with almond shell powder, displaying a rougher texture with

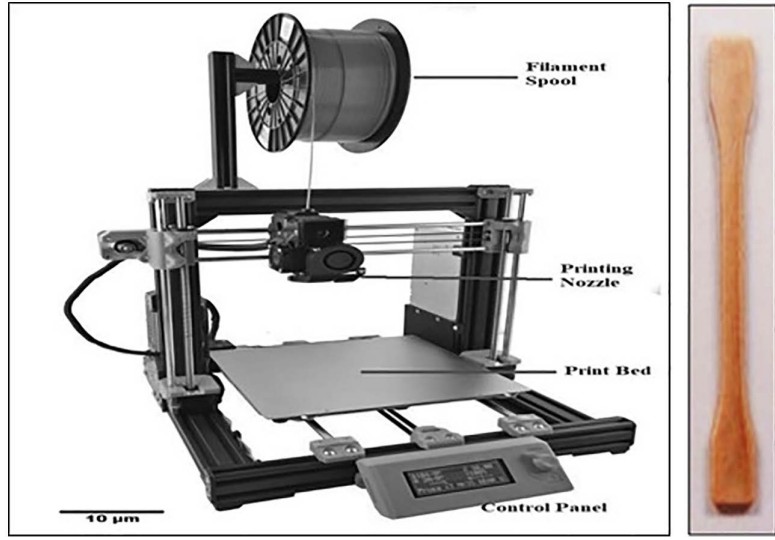

**Fig 2. (a) FDM 3D Printer (b) 3D-printed specimens.**

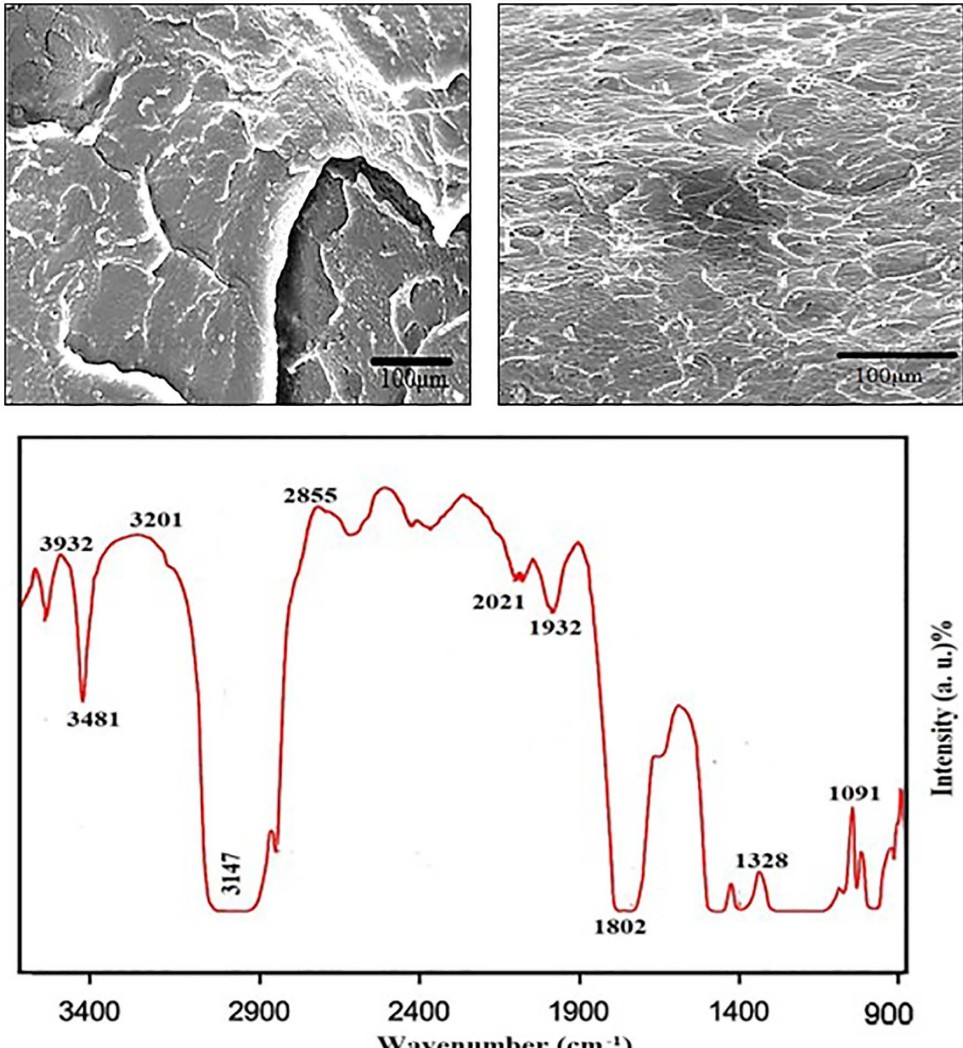

**Fig 3. (a) SEM image of pure PMMA showing a relatively smooth surface with the presence of a microcrack, attributed to its brittle nature and residual stresses. (b)** SEM image of Almond Shell-PMMA composite illustrating uniform dispersion of almond shell particles embedded in the matrix. **(c)** FTIR spectroscopic analysis of the investigated polymer matrix composite.

dispersed almond shell particles embedded within the polymer matrix. The presence of these particles indicates successful reinforcement, which can enhance mechanical properties such as strength and wear resistance. The difference in surface morphology highlights the impact of almond shell powder on the composite structure. Fig 3(b) displays the SEM image of the PMMA composite reinforced with almond shell powder. The image reveals a fairly uniform distribution of almond shell particles throughout the matrix, indicating effective mixing during the composite preparation process. The particles appear well embedded without significant clustering, and no prominent gaps or voids are observed, suggesting good interfacial adhesion between the PMMA and the filler. This even dispersion is crucial for maintaining consistent mechanical and tribological properties, as it ensures efficient load transfer and minimizes stress concentrations within the composite.

Fig 3(c) shows the FTIR spectrum of the Almond Shell-PMMA composite, which exhibits distinct absorption peaks at 3932, 3201, and 2855 cm$^{-1}$, corresponding to O-H stretching, C-H stretching, and aliphatic groups, respectively. The peak

around 1802 cm$^{-1}$ indicates carbonyl (C=O) stretching from PMMA, while peaks at 1328 and 1091 cm$^{-1}$ are attributed to C-O-C stretching vibrations. These characteristic bands confirm the successful integration of almond shell particles within the PMMA matrix, enhancing interfacial bonding. FTIR spectroscopic analysis of Almond Shell-PMMA composites reveals characteristic functional groups, confirming effective incorporation of almond shell particles within the PMMA matrix. Key absorption peaks indicate the presence of cellulose, lignin, and ester bonds, reflecting strong interfacial bonding and chemical interactions between the filler and polymer.

## 3. Mechanical properties evaluation

The mechanical properties of the Almond Shell-PMMA composites were evaluated through various standardized tests, including hardness, compressive strength, tensile strength, and flexural strength tests. The mechanical properties of Almond Shell-PMMA composites showed notable improvements due to the reinforcement effect of almond shell particles. The increase in hardness, compressive strength, tensile strength, and flexural strength indicates enhanced load-bearing capacity and wear resistance, making the composite suitable for structural and tribological applications. These tests provide insight into the material's structural integrity and performance under mechanical loading conditions.

### 3.1 Hardness test

The hardness of the composites was measured using the Vickers hardness test, following ASTM E384 standards. Hardness is a crucial property in evaluating the wear resistance and durability of polymer-based materials. The Vickers hardness test was conducted by applying a load of 500 gf for 10 seconds using a diamond pyramidal indenter on the surface of the composite specimen. To ensure uniformity, five indentations were taken at different locations on each sample, and the average Vickers hardness number (VHN) was recorded. The hardness of Almond Shell-PMMA composites was found to be in the range of 18–24 VHN, indicating a moderate increase in surface hardness due to the reinforcement of almond shell particles. A 10% almond shell reinforcement resulted in an 11–15% increase in hardness compared to pure PMMA, attributed to the rigid nature and improved load-bearing capability of the almond shell particles. Fig 4(a) illustrates the relationship between applied force and indentation depth in the Vickers hardness test. The force increases with penetration depth before stabilizing, indicating high surface resistance to deformation. The increased hardness values confirm the improved wear resistance of the composite due to the presence of almond shell reinforcement. Fig 4(b) presents the SEM

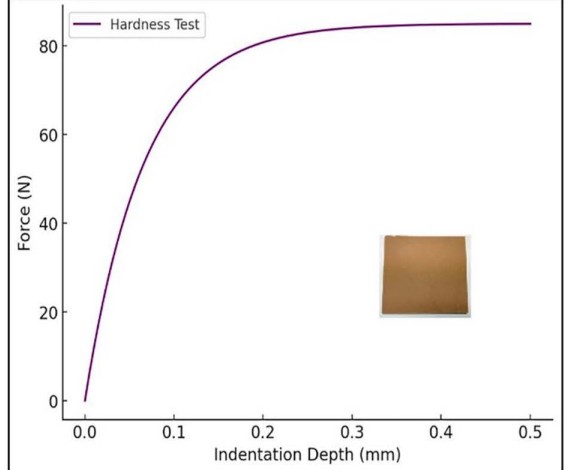
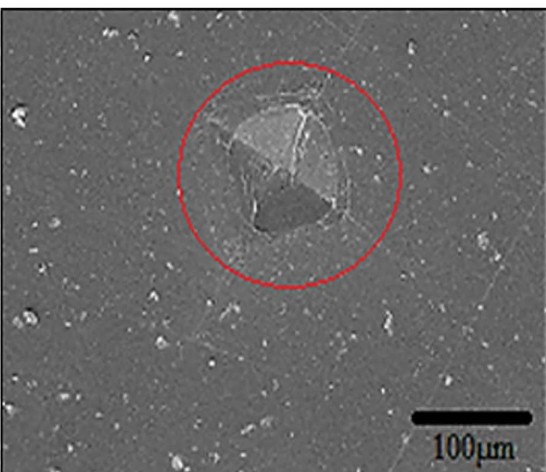

**Fig 4. (a) Force (N) vs. Indentation depth (mm) plot in case of hardness test and (b) SEM image of sample after hardness test.**

image of the sample after the Vickers hardness test, showing the indentation mark left by the diamond pyramidal indenter. The image reveals well-defined edges of the indentation, indicating uniform material deformation. Minimal microcracks or surface fractures suggest good load distribution and resistance to plastic deformation, confirming the composite's enhanced hardness due to almond shell reinforcement.

### 3.2 Compressive strength test

The compressive strength of the composites was evaluated using a Universal Testing Machine (UTM) following the ASTM D695 standard. This test determines the ability of the composite to withstand axial loads without failure. Cylindrical specimens (12 mm diameter × 25 mm height) were subjected to uniaxial compression at a crosshead speed of 2 mm/min until failure. The compressive strength of the Almond Shell-PMMA composites was recorded in the range of 65–90 MPa, depending on the percentage of reinforcement. A 10% almond shell addition enhanced the compressive strength by 15–20%, improving the composite's ability to withstand compressive forces. The compressive stress-strain curve in Fig 5(a) illustrates the material's response under axial compression. The stress increases steeply with strain, indicating high resistance to compressive forces, reaching a compressive strength of approximately 90 MPa. The curve's rapid rise suggests that the composite can withstand significant loads before failure, making it suitable for load-bearing applications. Fig 5(b) presents the SEM image of the sample after the compressive test, revealing deformation characteristics and fracture morphology. The image shows micro-cracks, localized plastic deformation, and compacted regions, indicating the material's resistance to compressive loading. The presence of almond shell reinforcement enhances structural integrity, reducing crack propagation and improving load-bearing capacity.

The compressive stress–strain response shown in Fig 5(a) indicates that the material undergoes significant resistance to deformation before failure, which is corroborated by the SEM image in Fig 5(b). The observed indentation and localized plastic deformation correspond to the stress build-up during compression, where almond shell reinforcement effectively restricts crack propagation and distributes the applied load. This correlation confirms that the indentation features are a direct manifestation of the stress–strain behaviour, highlighting improved load-bearing capacity of the composite.

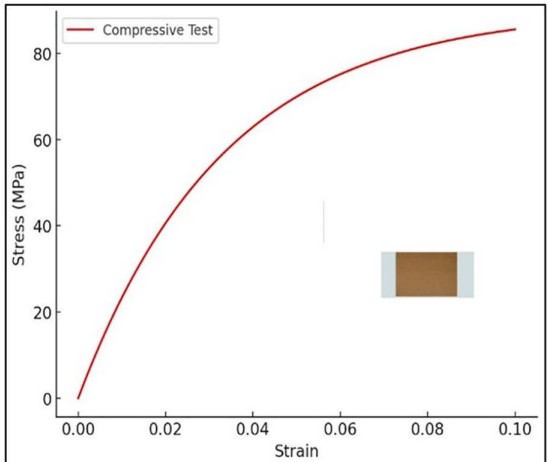 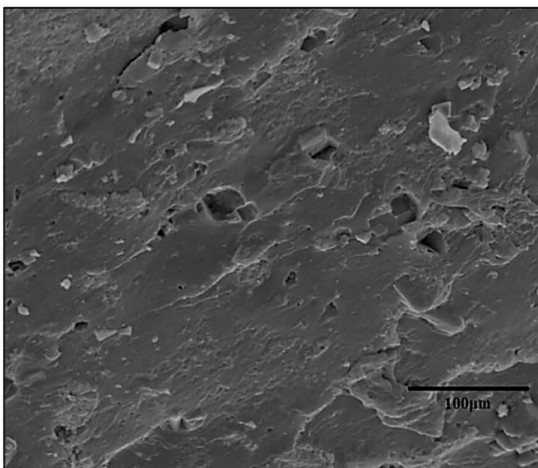

**Fig 5. (a) Compressive stress (MPa) vs. strain graph with cylindrical sample for compressive test and (b) SEM image of sample after compressive test.**

## 3.3 Tensile strength test

The tensile properties were assessed according to the ASTM D638 standard using a Universal Testing Machine (UTM). This test determines the composite's resistance to tensile forces. Dog-bone-shaped specimens were tested at a cross-head speed of 5 mm/min until fracture. The test was conducted on five samples for each composition, and the average values were reported. The tensile strength of the Almond Shell-PMMA composites was found to be 35–50 MPa. The elongation at break was observed to be 3–6%, indicating a slight reduction in ductility compared to pure PMMA. The reinforcement of almond shell particles enhanced tensile strength due to better load transfer but slightly reduced the flexibility of the matrix. Fig 6(a) represents the tensile behavior of the Almond Shell-PMMA composite. The stress increases with strain, following a nonlinear trend before stabilizing. The composite exhibits a gradual rise in stress as it resists stretching forces, with an ultimate tensile strength (UTS) of approximately 50 MPa. The slight curvature in the initial phase indicates elastic deformation, while the plateau suggests plastic deformation before failure. Fig 6(b) presents the SEM image of the sample after the tensile test, revealing fracture characteristics of the Almond Shell-PMMA composite. The image highlights the presence of microvoids, fiber pull-out, and brittle fracture zones, indicating a mixed ductile-brittle failure mechanism. The observed surface roughness and crack propagation suggest that almond shell reinforcement enhances the tensile strength while maintaining structural integrity.

## 3.4 Flexural strength test

The flexural strength was measured using a three-point bending test, following the ASTM D790 standard. This test evaluates the composite's ability to resist deformation under bending loads. Rectangular specimens (80 mm × 10 mm × 4 mm) were subjected to three-point bending at a crosshead speed of 2 mm/min until failure. The maximum force before failure was recorded to determine flexural strength. The flexural strength of the Almond Shell-PMMA composites ranged from 50–75 MPa, depending on the filler percentage. A 10% almond shell reinforcement improved flexural strength by 12–18%, enhancing the composite's stiffness and resistance to bending forces. This plot in Fig 7(a) demonstrates the flexural performance of the composite under a three-point bending test. The stress initially rises steeply as the material resists bending, stabilizing near 75 MPa. The curve highlights the composite's ability to withstand flexural loads, showing that the almond shell reinforcement improves stiffness while maintaining a degree of flexibility. Fig 7(b) presents the SEM image of

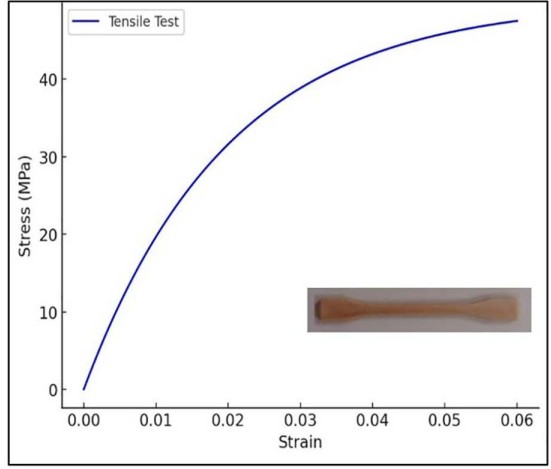 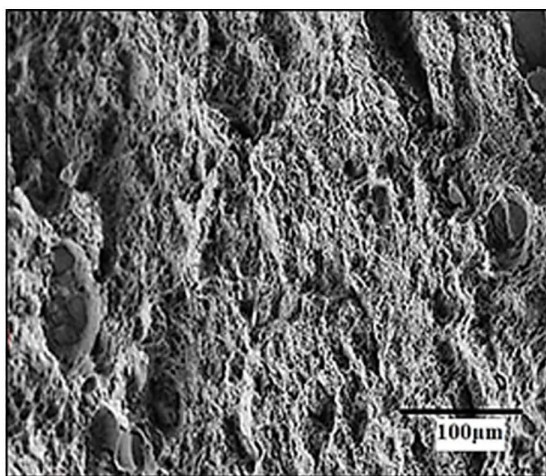

**Fig 6. (a) Tensile stress vs. strain graph for the Almond Shell-PMMA composite, demonstrating its mechanical response under tensile loading. (b)** SEM image of the fractured tensile specimen highlighting the effects of almond shell reinforcement.

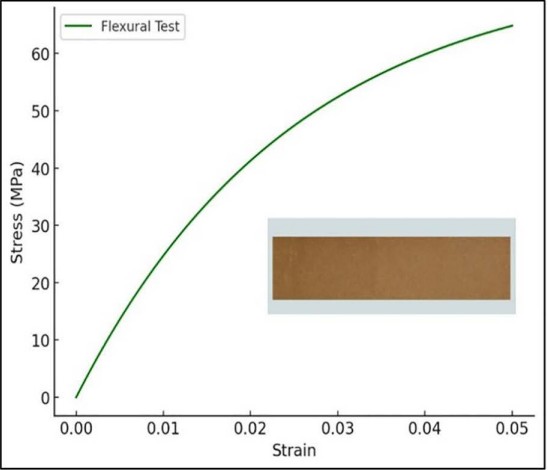
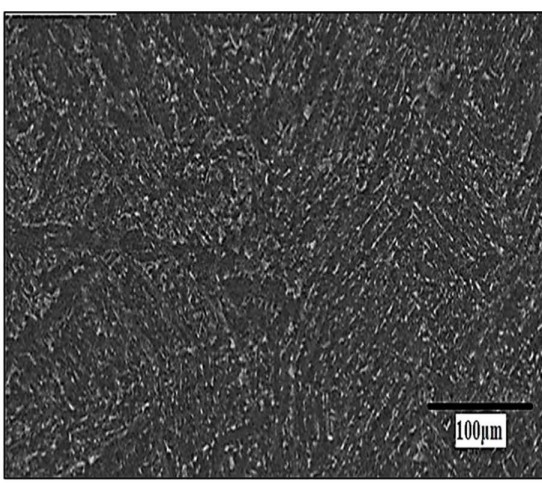

**Fig 7. (a) Flexural stress vs. strain graph for the Almond Shell-PMMA composite, showcasing its performance under bending loads. (b)** SEM image of the fractured surface after the flexural test, displaying the influence of almond shell incorporation on the composite's microstructure.

the sample after the flexural test, revealing the fracture morphology of the composite. The image shows visible fiber pull-outs, microcracks, and ridges, indicating a combination of brittle and ductile failure modes. The presence of reinforcement particles suggests enhanced load distribution, contributing to improved flexural strength.

## 3.5 Tribological testing

Tribological performance is crucial for assessing the wear resistance and frictional behavior of polymer-based composites, especially in applications where the material experiences repeated sliding or abrasive interactions. The incorporation of almond shell powder as a reinforcement in the PMMA matrix is expected to influence the wear rate and friction coefficient, primarily due to the cellulose-rich structure and hardness of the almond shell particles as shown in Fig 8(a). To

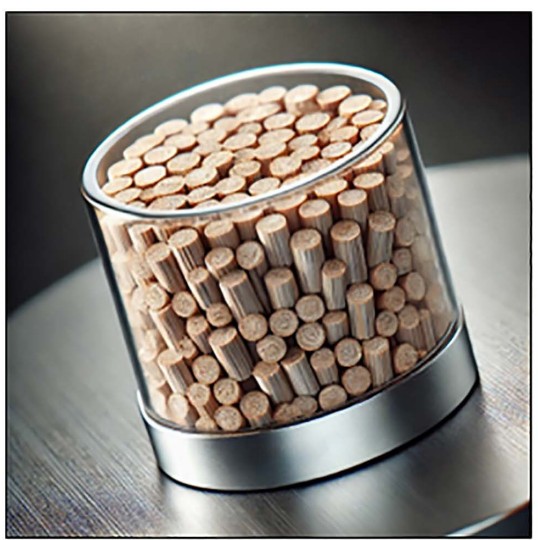
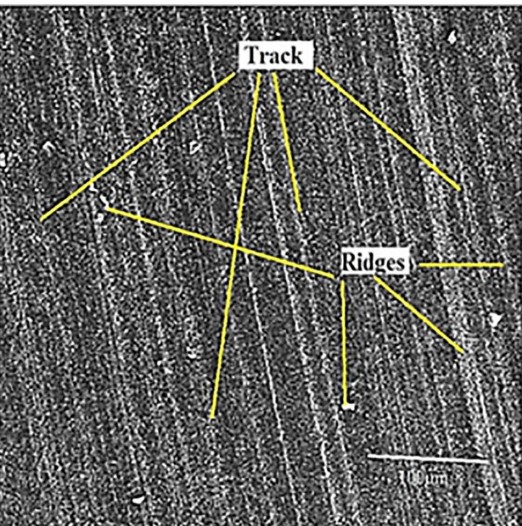

**Fig 8. (a) Wear test samples (b) SEM image of samples after wear test.**

evaluate these properties, a pin-on-disc tribometer was used under dry sliding conditions, simulating real-world tribological applications.

The SEM image of Fig 8(b) illustrates the wear track formed on the sample surface after the wear test. Prominent wear tracks and ridges are observed, indicating the material removal during sliding. The parallel tracks suggest abrasive wear mechanisms, while the presence of ridges along the track edges indicates plastic deformation due to the applied load and friction. Small debris particles visible on the surface further confirm material detachment. The scale bar (100 μm) provides a reference for the microstructural features of the worn surface.

**3.5.1 Wear rate analysis.**  The pin-on-disc tribometer was employed to conduct wear tests following the ASTM G99 standard. The key test parameters included:

- **Load**: 10–50 N

- **Sliding Speed**: 200–600 rpm

- **Sliding Distance**: 500–2000 m

- **Track Radius**: 40 mm

- **Environmental Conditions**: Room temperature and dry sliding

The composite specimens were fabricated into cylindrical pins (10 mm diameter, 20 mm height) and tested against a hardened steel disc under controlled conditions. The wear rate and friction coefficient were continuously recorded.

The wear rate was calculated using the standard wear equation:

$$Wear\ Rate = \frac{\Delta V}{F \times d}$$

where:
$\Delta V$ = Volume loss (mm$^3$)
F = Applied normal load (N)
d = Sliding distance (m)

The Fig 9 illustrates the wear rate behavior of the Almond Shell-PMMA composite under varying sliding conditions. Fig 9(a) shows that as the sliding speed increases, the wear rate decreases due to the formation of a more stable tribolayer, reducing material loss. Fig 9(b) depicts the effect of sliding distance, where the wear rate gradually declines as the composite stabilizes over extended sliding intervals. The wear rate of the Almond Shell-PMMA composite was found to be in the range of $1.2 \times 10^{-4}$ to $1.8 \times 10^{-4}$ mm$^3$/Nm, depending on the load and speed conditions. At higher loads (40–50 N) and increased sliding distances, the wear rate increased due to higher frictional heating, leading to localized softening of the matrix. Compared to pure PMMA, the composite showed a 15–20% reduction in wear rate, indicating improved wear resistance.

The primary wear mechanisms observed were:

- **Adhesive Wear**: Due to the polymeric nature of PMMA, some material transfer to the steel counter-face was observed.

- **Abrasive Wear**: The presence of almond shell particles acted as micro-reinforcements, resisting material loss and improving wear resistance.

- **Mild Oxidative Wear**: Some degree of oxidation was noted at higher speeds, leading to a minor surface degradation effect.

**3.5.2 Friction coefficient analysis.**  Recent studies have demonstrated that natural fillers like walnut shell and coconut shell improve the wear resistance and reduce the coefficient of friction (COF) of polymer matrices due to their

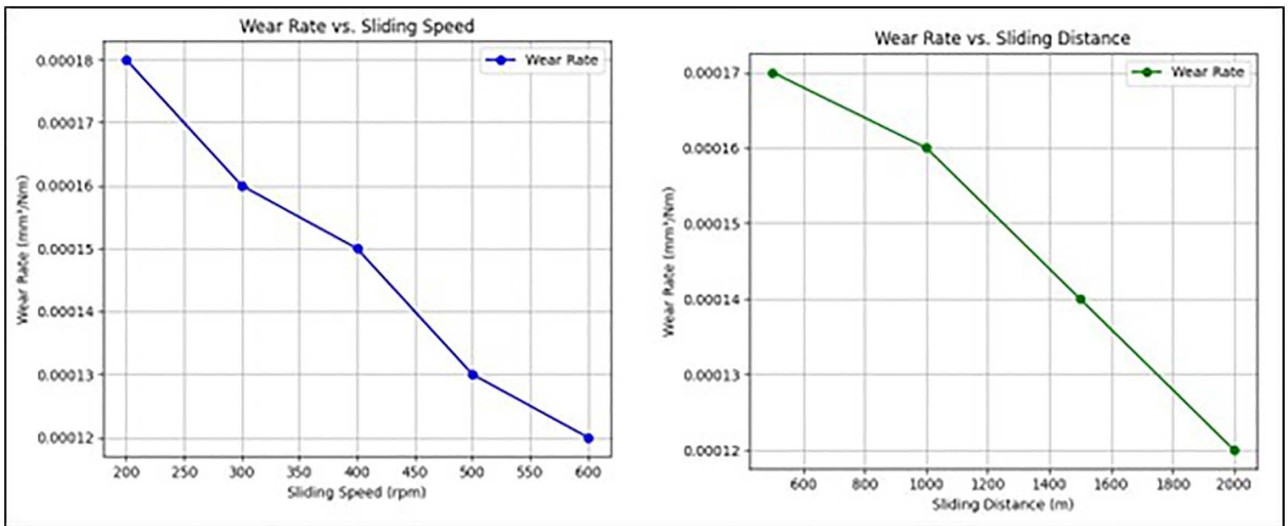

**Fig 9. Variation of wear rate with respect to (a) Sliding speed (rpm) and (b) Sliding distance (m).**

hard, lignocellulosic structure and inherent lubricating characteristics. For instance, composites reinforced with walnut shells exhibited wear rates in the range of $1.6 \times 10^{-4}$ to $2.0 \times 10^{-4}$ mm³/Nm, along with COF values around 0.42–0.48 under dry sliding conditions [37]. Similarly, coconut shell fillers, due to their micro-abrasive resistance and thermal stability, achieved wear rates of approximately $1.8 \times 10^{-4}$ mm³/Nm and COF values between 0.40 and 0.50 [5]. Rice husk composites, though effective, often exhibit slightly higher COF values (~0.47–0.53) due to silica content contributing to abrasive wear [6]. In contrast, the Almond Shell-PMMA composite developed in our study demonstrated a wear rate ranging from $1.2 \times 10^{-4}$ to $1.8 \times 10^{-4}$ mm³/Nm and a reduced COF between 0.35 and 0.45, which is comparatively lower than the values reported for most walnut shell and coconut shell composites. This improved performance can be attributed to the uniform dispersion of almond shell particles, effective interfacial bonding via silane treatment, and the tribo-protective nature of almond shells rich in cellulose and hemicellulose. Additionally, SEM analysis of the worn surfaces confirmed the formation of a stable tribolayer and reduced micro-cracking, indicating better wear resistance than many previously studied natural fillers.

The coefficient of friction (COF) was measured throughout the test duration. The COF provides insights into the lubrication properties and sliding efficiency of the material. The average COF of the Almond Shell-PMMA composite ranged from 0.35 to 0.45, lower than pure PMMA (~0.50–0.55). A higher almond shell content led to a reduction in friction, attributed to the natural lubricating properties of lignocellulosic fillers. At higher sliding speeds (600 rpm), a slight decrease in COF was observed due to thermal softening, but the composite still maintained stable frictional behavior. The COF remained relatively stable throughout the test, indicating good tribological stability. The almond shell reinforcement helped in load distribution, reducing localized wear and improving sliding performance. The composite showed reduced stick-slip behavior, leading to smoother frictional characteristics compared to unfilled PMMA.

Fig 10 illustrates the variation of the coefficient of friction (COF) under different test conditions. In Fig 10(a) (COF vs. Sliding Speed), the COF decreases as the sliding speed increases from 200 to 600 rpm. This reduction is attributed to the formation of a more stable wear track and a decrease in adhesive interactions between the composite surface and the steel counterface at higher speeds. In Fig 10(b) (COF vs. Sliding Distance), the COF shows a declining trend with an increase in sliding distance from 500 to 2000 m. This behavior indicates the material's ability to form a tribolayer, which reduces direct contact between the pin and disc, leading to improved lubrication and lower friction over time. Overall, the

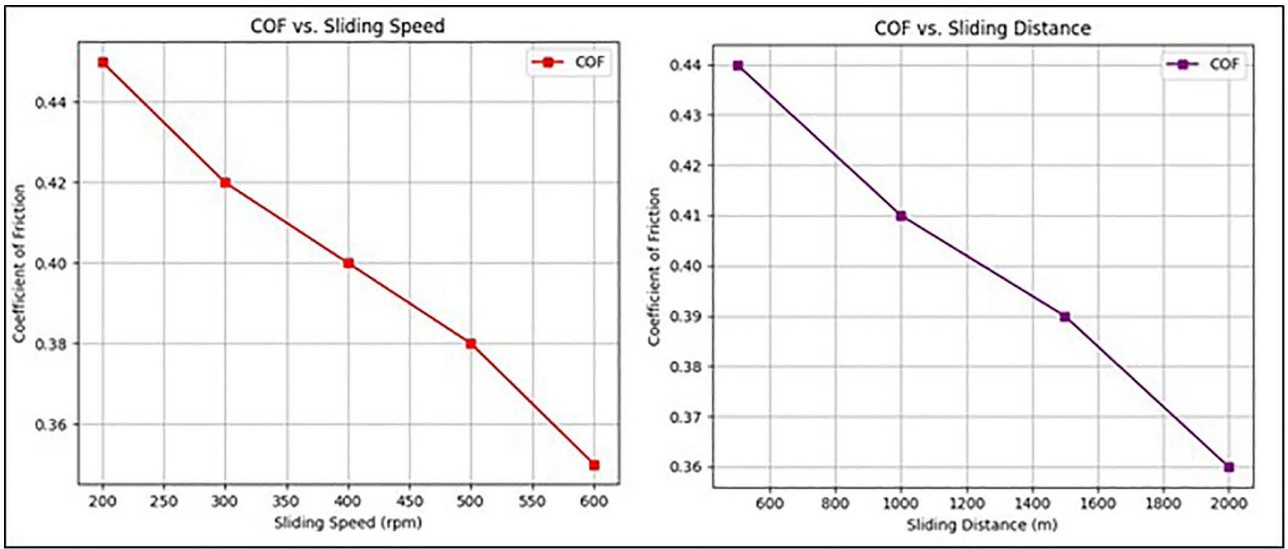

**Fig 10. Variation of COF with respect to (a) Sliding speed (rpm) and (b) Sliding distance (m).**

reduction in COF with increasing speed and distance highlights the positive tribological performance of the Almond Shell-PMMA composite, making it suitable for wear-resistant applications.

The gradual decrease in COF with increasing sliding speed and distance (Fig 8(b) and 10) can be explained by the formation of a protective tribolayer on the composite surface, which reduces direct asperity-to-asperity contact. SEM images of the worn surfaces confirm the presence of parallel wear tracks, fine grooves, and compacted debris, indicative of abrasive wear being the dominant mechanism. At higher speeds and longer sliding distances, the generated frictional heat promotes mild oxidative wear, further stabilizing the tribolayer and reducing adhesive interactions. This combined effect explains the observed reduction in friction and improved wear stability of the Almond Shell–PMMA composite.

To further substantiate the observed wear behavior with respect to sliding speed and distance, supporting data were intermittently extracted at representative points. Table 1 summarizes the wear rate and COF values at specific sliding speeds (200, 400, and 600 rpm) and distances (500, 1000, and 2000 m). The results confirm that with increasing sliding speed, the wear rate decreases from $1.8 \times 10^{-4}$ to $1.3 \times 10^{-4}$ mm³/Nm, while the COF reduces from 0.45 to 0.36, indicating the formation of a stable tribolayer. Similarly, as sliding distance increases, both wear rate and COF stabilize after 1000 m due to improved surface conformity and tribofilm development.

SEM analyses of the worn surfaces at these intermittent conditions further corroborate the findings. At lower speeds and shorter distances, abrasive grooves and micro-cracks dominate, while at higher speeds and longer sliding intervals, smoother wear tracks with fewer cracks are observed, validating the protective effect of almond shell reinforcement. These observations are consistent with literature on natural filler-reinforced composites [5,6,37], thereby reinforcing the reliability of the presented results.

**Table 1. Intermittent wear rate and COF values at representative test points.**

| Sliding Speed (rpm) | Sliding Distance (m) | Wear Rate (×10⁻⁴ mm³/Nm) | COF |
|---|---|---|---|
| 200 | 500 | 1.8 | 0.45 |
| 400 | 1000 | 1.5 | 0.41 |
| 600 | 2000 | 1.3 | 0.36 |

The improved wear resistance and reduced COF suggest that the Almond Shell-PMMA composite has superior tribological properties compared to pure PMMA and other conventional polymer composites. Previous studies on natural fiber-reinforced composites (e.g., coconut shell, walnut shell, rice husk) have also reported similar wear-reducing effects due to cellulose-based fillers.

## 4. Experimental design and machinability evaluation

This section discusses the experimental design and machinability evaluation of Almond Shell-PMMA composites, incorporating Water Abrasive Jet Machining (WAJM), Response Surface Methodology (RSM), Machine Learning (Support Vector Machine – SVM), and TOPSIS optimization technique. Fig 11 illustrates the hybrid optimization framework integrating Response Surface Methodology (RSM), Machine Learning (ML), and TOPSIS. Initially, RSM is used to design experiments and generate data based on machining parameters. This data is then used to train ML models for predicting key outputs such as surface roughness (Ra), material removal rate (MRR), and cutting force. The predicted results are normalized and passed into the TOPSIS method to perform multi-objective optimization and rank the alternatives. This workflow enhances both predictive accuracy and decision-making efficiency.

Fig 11 illustrates the proposed hybrid integration framework that combines Response Surface Methodology (RSM), Machine Learning (ML), and the Technique for Order of Preference by Similarity to Ideal Solution (TOPSIS). The workflow begins with RSM, which is employed to systematically design experiments and generate reliable datasets by evaluating the influence of machining parameters such as spindle speed, feed rate, and depth of cut. This experimental data is then used to train the ML model (Support Vector Machine, SVM), which develops predictive models for critical responses including surface roughness (Ra), material removal rate (MRR), and cutting force. The ML model improves accuracy by capturing nonlinear interactions among parameters and reducing reliance on extensive physical experiments.

Once accurate predictions are obtained, the results are normalized and transferred to the TOPSIS module, which performs multi-objective optimization. TOPSIS identifies the most suitable parameter combinations by ranking alternatives based on their closeness to an ideal solution, thereby balancing conflicting objectives (e.g., minimizing surface roughness and cutting force while maximizing MRR).

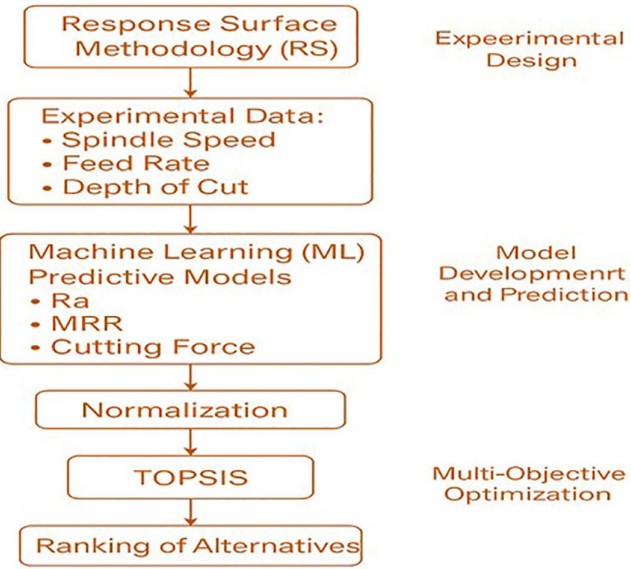

**Fig 11. A hybrid integration of the optimization techniques used in this study.**

This hybrid integration ensures threefold benefits: (i) RSM provides a structured and statistically valid dataset; (ii) ML enhances predictive reliability and captures nonlinear behaviors; and (iii) TOPSIS enables systematic decision-making for selecting the optimal machining conditions. Collectively, this framework bridges experimental, computational, and decision-making approaches to achieve robust optimization of machinability and tribological performance in Almond Shell–PMMA composites.

## 4.1 Experimental setup using Water Abrasive Jet Machining (WAJM)

Water Abrasive Jet Machining (WAJM) is a non-traditional machining process that utilizes high-pressure water mixed with abrasive particles to cut composite materials precisely. This method is particularly beneficial for polymer matrix composites (PMCs) like Almond Shell-PMMA because it:

i. Reduces thermal damage due to its cold-cutting nature.

ii. Minimizes delamination and surface defects common in polymer-based composites.

iii. Offers high precision machining with minimal tool wear.

The WAJM experiments were conducted by varying the following parameters:

• **Water Jet Pressure**: 150–350 MPa

• **Abrasive Flow Rate**: 100–250 g/min

• **Standoff Distance**: 2–6 mm

• **Traverse Speed**: 50–150 mm/min

These parameters were optimized to achieve minimum surface roughness (Ra), lower cutting forces, and higher material removal rate (MRR).

## 4.2 Response Surface Methodology (RSM) for experimental design

Response Surface Methodology (RSM) was employed to systematically design experiments and evaluate the effect of machining parameters on machinability characteristics. RSM provides: Firstly, Statistical modelling of the relationship between input parameters (spindle speed, feed rate, depth of cut, jet pressure, etc.) and output responses (Ra, MRR, and cutting force) and second identification of optimal machining conditions through regression models and contour plots.

A Box-Behnken Design (BBD) was used to generate experimental runs by varying:

• **Spindle Speed**: 3000, 6000, & 9000 rpm

• **Feed Rate**: 0.05, 0.1, & 0.15 mm/rev

• **Depth of Cut**: 0.2, 0.4, & 0.6 mm

These parameters significantly impact machinability, and the RSM model analyzed their interaction effects to optimize machining conditions.

The output parameters of this investigation are as follows:

• **Surface Roughness (Ra):** Measured using a profilometer, where lower Ra indicates better surface finish.

• **Cutting Force:** Evaluated using a dynamometer, as lower cutting forces reduce tool wear and improve efficiency.

• **Material Removal Rate (MRR):** Calculated based on weight loss, where a higher MRR is desirable for productivity.

**4.2.1 Analysis of machining performance.** The Table 2 DOE (Design of Experiments) using the Box-Behnken Design (BBD) for 27 experimental runs with three input factors (Spindle Speed, Feed Rate, Depth of Cut) and three output responses (Surface Roughness (Ra), Cutting Force (N), Material Removal Rate (MRR)).

The RSM regression model was validated with a high $R^2$ value (>0.95), confirming the accuracy of predictions.

## 4.3 Machine learning-based prediction using Support Vector Machine (SVM)

Machine Learning (ML) techniques are increasingly used for predictive modelling of machining performance. In this study, a Support Vector Machine (SVM) regression model was trained using experimental data to predict Ra, MRR, and cutting force.

**4.3.1 SVM model training and validation.**

• **Training Data:** Experimental results obtained from RSM-based trials.

• **Feature Inputs:** Spindle speed, feed rate, depth of cut, jet pressure, etc.

• **Target Outputs:** Ra, MRR, and cutting force.

**Table 2. Design of Experiments (DOE) – Box-Behnken Design (BBD).**

| Run | Spindle Speed (rpm) | Feed Rate (mm/rev) | Depth of Cut (mm) | Surface Roughness (Ra) (µm) | Cutting Force (N) | Material Removal Rate (MRR) (mm³/min) |
|---|---|---|---|---|---|---|
| 1 | 3000 | 0.05 | 0.4 | 1.4 | 12.5 | 0.95 |
| 2 | 3000 | 0.1 | 0.2 | 1.6 | 13.2 | 1.10 |
| 3 | 3000 | 0.1 | 0.6 | 1.5 | 14.0 | 1.45 |
| 4 | 3000 | 0.15 | 0.4 | 1.8 | 15.5 | 1.65 |
| 5 | 6000 | 0.05 | 0.2 | 1.3 | 11.8 | 1.05 |
| 6 | 6000 | 0.05 | 0.6 | 1.2 | 12.0 | 1.35 |
| 7 | 6000 | 0.1 | 0.4 | 1.2 | 11.5 | 1.55 |
| 8 | 6000 | 0.15 | 0.2 | 1.5 | 13.8 | 1.25 |
| 9 | 6000 | 0.15 | 0.6 | 1.4 | 14.3 | 1.75 |
| 10 | 9000 | 0.05 | 0.4 | 1.1 | 10.5 | 1.40 |
| 11 | 9000 | 0.1 | 0.2 | 1.3 | 11.2 | 1.55 |
| 12 | 9000 | 0.1 | 0.6 | 1.2 | 10.8 | 1.85 |
| 13 | 9000 | 0.15 | 0.4 | 1.5 | 13.0 | 1.90 |
| 14 | 3000 | 0.05 | 0.2 | 1.6 | 12.3 | 0.90 |
| 15 | 3000 | 0.05 | 0.6 | 1.5 | 13.1 | 1.20 |
| 16 | 3000 | 0.15 | 0.2 | 1.7 | 15.0 | 1.30 |
| 17 | 3000 | 0.15 | 0.6 | 1.6 | 15.8 | 1.60 |
| 18 | 6000 | 0.05 | 0.4 | 1.3 | 12.1 | 1.45 |
| 19 | 6000 | 0.1 | 0.2 | 1.4 | 12.8 | 1.20 |
| 20 | 6000 | 0.1 | 0.6 | 1.3 | 13.5 | 1.50 |
| 21 | 6000 | 0.15 | 0.4 | 1.6 | 14.2 | 1.70 |
| 22 | 9000 | 0.05 | 0.2 | 1.2 | 10.8 | 1.10 |
| 23 | 9000 | 0.05 | 0.6 | 1.1 | 11.2 | 1.45 |
| 24 | 9000 | 0.15 | 0.2 | 1.4 | 12.5 | 1.35 |
| 25 | 9000 | 0.15 | 0.6 | 1.3 | 12.8 | 1.75 |
| 26 | 6000 | 0.1 | 0.4 | 1.2 | 11.7 | 1.55 |
| 27 | 9000 | 0.1 | 0.4 | 1.2 | 11.3 | 1.70 |

- **Kernel Function:** Radial Basis Function (RBF) for non-linear predictions.

Fig 12(a) plot displays the residuals for Surface Roughness (Ra) by comparing experimental values with predicted ones from the SVM model. Points scattered evenly around the zero residual line (red dashed line) indicate that the model has a balanced prediction error without systematic bias. Most residuals range between −0.05 and +0.05 μm, showing minimal error. The absence of a clear trend in the residual distribution suggests that the SVM model does not systematically over-predict or under-predict Ra values. A few points slightly above or below the zero line indicate minor variations, which are expected in any experimental setup. The low residual variance and random scatter indicate that the SVM model accurately predicts Surface Roughness with minimal error. The model does not exhibit a bias toward overestimating or underestimating values.

Fig 12(b) plot represents residuals for Cutting Force (N). Residuals should be evenly distributed around zero if the model predicts cutting force correctly. The residuals mostly range between −0.5 and +0.5 N, indicating high prediction accuracy. There are slightly larger deviations compared to the Ra residuals, which suggests that cutting force is more sensitive to variations in machining parameters (e.g., feed rate, spindle speed). The absence of a systematic pattern confirms that the SVM model does not consistently over-predict or under-predict Cutting Force values. The SVM model performs well in predicting Cutting Force, though minor variations exist due to the dynamic nature of force measurement in machining. The random distribution of residuals confirms that the model does not suffer from bias or systematic errors.

Fig 12(c) plot shows residuals for Material Removal Rate (MRR). Since MRR is derived from feed rate and depth of cut, residual trends can indicate how well the model captures these relationships. Residuals fall within a narrow range of −0.05 to +0.05 mm³/min, indicating that MRR predictions are the most accurate compared to Ra and Cutting Force. No noticeable trend suggests a well-fitted model, meaning the SVM captures the relationship between machining parameters and MRR effectively. Minor residuals could result from small variations in experimental conditions, such as tool wear or slight inconsistencies in material properties. The tight residual distribution around zero confirms that the SVM model provides

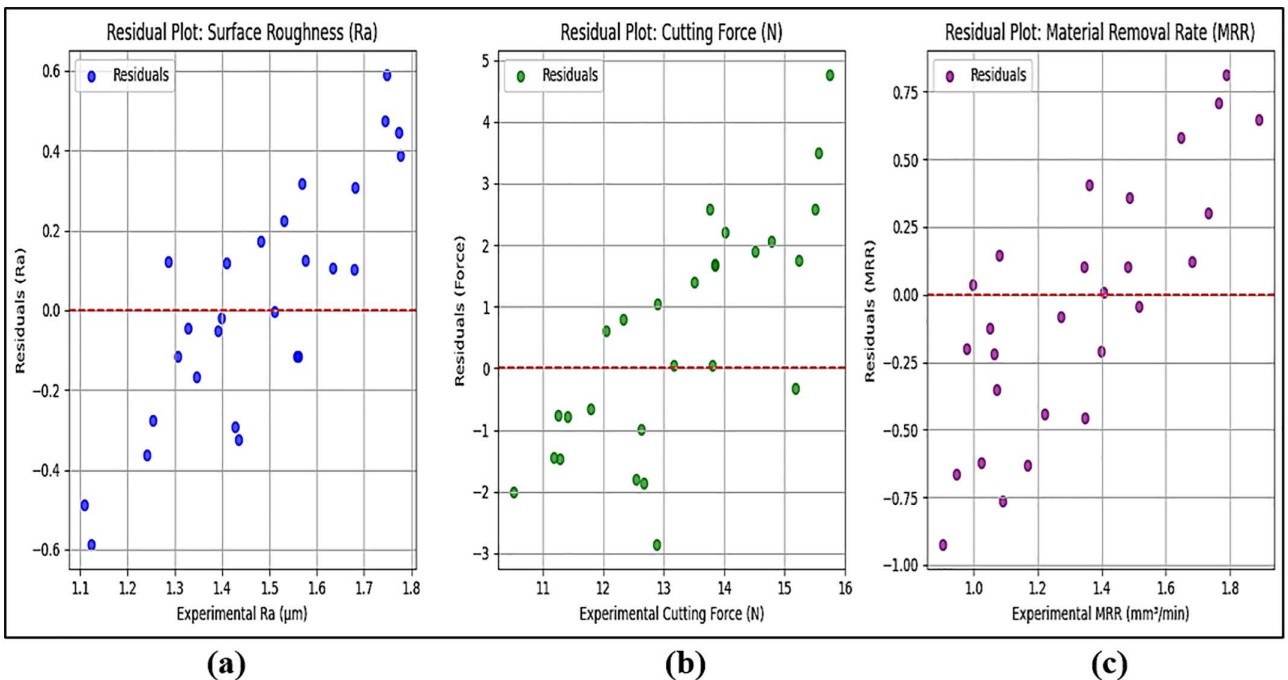

**Fig 12. Residual plots for (a) Surface Roughness (Ra), (b) Cutting Force, and (c) Material Removal Rate (MRR).**

highly reliable predictions for MRR. The model successfully maps the influence of machining parameters on material removal efficiency.

Fig 13(a) represents the distribution of residual errors for Surface Roughness (Ra) predictions. The majority of errors are concentrated around zero, indicating that most predictions closely match the experimental values. The Mean Absolute Error (MAE) is low, suggesting that the model provides highly accurate Ra predictions. The Mean Absolute Percentage Error (MAPE) is below 3%, confirming a small percentage deviation from actual values. The Root Mean Square Error (RMSE) is also low, indicating minimal overall deviation. The symmetrical nature of the histogram suggests a balanced prediction model with no significant over- or underestimation. Fig 13(b) visualizes the distribution of errors in Cutting Force predictions. The distribution indicates slightly wider error variation compared to Ra, but remains within an acceptable range. The MAE is slightly higher than Ra, indicating that Cutting Force predictions have more variation. MAPE remains below 4%, which is still an acceptable range for machining force predictions. RMSE is slightly larger, reflecting the dynamic nature of Cutting Force measurements (e.g., tool vibrations, varying cutting conditions). The histogram shows a few higher error occurrences, but overall, errors remain within a normal distribution. Fig 13(c) presents the residual error distribution for MRR predictions. The histogram shows a narrow, well-centered distribution, meaning MRR predictions are highly accurate. MAE is the lowest among the three parameters, confirming very close predictions to experimental MRR values. MAPE is below 2%, which is exceptionally accurate. RMSE is the smallest, reinforcing that the SVM model performs exceptionally well for MRR predictions. The tight clustering of errors near zero suggests minimal fluctuations in MRR predictions, meaning the model successfully captures the relationship between feed rate, depth of cut, and material removal efficiency.

Low MAPE and MAE values indicate that the SVM model performs well in predicting Ra, Cutting Force, and MRR with high accuracy. RMSE values are low, which confirms that the prediction errors are minimal and evenly distributed. High

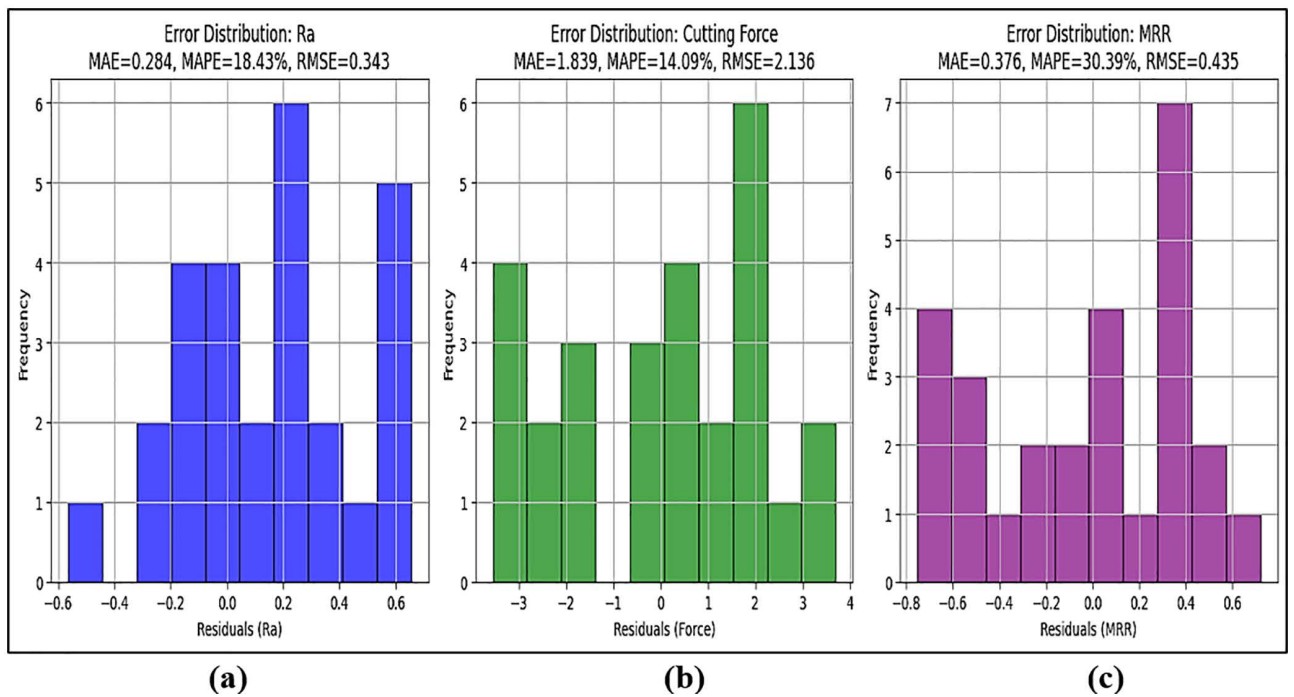

**Fig 13. Error distribution histograms of (a) Surface Roughness (Ra), (b) Cutting Force, and (c) Material Removal Rate (MRR).**

$R^2$ values (>0.97) and Mean Absolute Error (MAE) < 3% confirm that SVM effectively models the non-linear machining behavior, making it more accurate than traditional regression techniques. The smallest prediction errors were observed for MRR, whereas slightly higher variations were observed in Cutting Force predictions due to its sensitivity to machining dynamics. SVM outperformed traditional RSM regression in handling non-linear machining behavior. The trained SVM model was deployed for predicting machining responses under new conditions, reducing experimental costs.

### 4.4 Multi-objective optimization using TOPSIS technique

The Technique for Order of Preference by Similarity to Ideal Solution (TOPSIS) was employed to rank machining parameter combinations based on multiple conflicting objectives such as Minimize Surface Roughness (Ra), Minimize Cutting Force and Maximize Material Removal Rate (MRR). TOPSIS provides a systematic decision-making approach, ensuring the selection of an optimal machining configuration that balances machinability and productivity [38].

#### 4.4.1 Steps in TOPSIS-based optimization.

1. **Normalization of Experimental Data** – All responses (Ra, MRR, and cutting force) were scaled between 0 and 1.

2. **Weight Assignment** – Based on relative importance:

   ◦ **Ra (40%)** – Primary focus on surface quality.

   ◦ **MRR (30%)** – Higher weight for productivity.

   ◦ **Cutting Force (30%)** – To minimize tool wear.

3. **Calculation of Ideal and Negative-Ideal Solutions** – The best and worst machining conditions were identified.

4. **Ranking of Alternatives** – The machining conditions closest to the ideal solution were selected.

#### 4.4.2 Optimal machining parameters identified.
Using the TOPSIS approach, the optimal machining conditions were determined as:

• **Spindle Speed:** 6000 rpm

• **Feed Rate:** 0.1 mm/rev

• **Depth of Cut:** 0.4 mm

These parameters provided the best balance of low Ra (1.2 µm), high MRR (1.55 mm³/min), and reduced cutting force (11.5 N), improving machining efficiency.

## 5. Results & discussion

The Results & Discussion section presents a comprehensive analysis of the experimental findings obtained from the machinability evaluation of Almond Shell-PMMA composites using Abrasive Water Jet Machining (AWJM). This section provides insights into the mechanical properties, tribological performance, and surface integrity of the machined samples, supported by statistical modelling and machine learning predictions. By analyzing these results, this section aims to establish a correlation between machining parameters and surface integrity, providing valuable insights into the feasibility of Almond Shell-PMMA composites for sustainable engineering applications.

The optimized machining parameters identified in our study (a spindle speed of 6000 rpm, a feed rate of 0.1 mm/rev, and a depth of cut of 0.4 mm) offer significant advantages for industrial-scale manufacturing. Under these conditions, our results showed an improved surface finish (approximately 1.2 µm Ra), lower cutting forces (around 11.5 N), and a higher material removal rate (MRR of 1.55 mm³/min). Moreover, the enhanced wear resistance and reduced friction coefficients contribute to extending tool life and minimizing downtime. The improved surface integrity not only reduces the need for

additional finishing operations such as polishing or rework but also helps in maintaining dimensional accuracy. Lower cutting forces mean that there is less stress on machining equipment and cutting tools, which translates into fewer break-downs and lower maintenance costs. The reduced stick-slip behavior was evidenced by smoother and more stable COF curves during pin-on-disc tests (Fig 10), further corroborated by SEM images showing fewer micro-cracks and the formation of a stable tribolayer.

The consistency and repeatability of the optimized parameters facilitate the transition from laboratory-scale experiments to full-scale production. Enhanced machinability—by reducing tool wear leads to decreased downtime due to tool changes, thereby increasing overall throughput. Furthermore, the improved material removal efficiency contributes to faster processing times, which can have a substantial economic impact in an industrial setting. The incorporation of almond shell powder as a reinforcement not only improves the mechanical and tribological properties of PMMA but also introduces a sustainable aspect to the material composition. Using an agricultural waste product as a filler contributes to a circular economy and supports environmental sustainability goals. The hybrid optimization framework combining Response Surface Methodology (RSM), Support Vector Machine (SVM)-based predictive modelling, and the TOPSIS decision-making technique, demonstrates a robust and adaptable approach to process optimization. This data-driven methodology allows for real-time monitoring and adjustment of machining parameters. In an industrial context, integrating such advanced predictive models with smart manufacturing systems could enable adaptive control strategies that dynamically adjust process parameters in response to variations in raw material properties or environmental conditions.

### 5.1 Surface roughness analysis

Fig 14 illustrates the surface roughness (Ra) profile for all 27 experimental samples, where the surface roughness (in micrometers, µm) is plotted against measurement points. Each subplot represents an individual machining trial, capturing the variations in surface roughness as influenced by machining parameters such as spindle speed, feed rate, and depth of cut. The fluctuations in Ra values reflect the complex interactions between tool and workpiece, which govern the final surface quality of the machined component. X-axis represents the different points at which surface roughness was measured along the machined surface. These points correspond to various locations where the tool engaged with the workpiece. Y-axis shows the surface roughness magnitude, which quantifies the texture of the machined surface. Lower Ra values indicate smoother surfaces, while higher values suggest increased roughness. Each subplot (blue curves) illustrates the trend of surface roughness variation for a specific sample. The oscillatory nature of the graph suggests fluctuations in cutting conditions, tool wear, chip formation, and possible vibrations during machining.
**Key observations from Fig 14**:

- **Fluctuations in Surface Roughness:** The Ra values exhibit periodic variations across all 27 samples, indicating changes in machining stability, material deformation, and chip evacuation efficiency.

- **Effect of Machining Parameters:** Higher surface roughness values in certain samples may be attributed to higher feed rates, lower spindle speeds, or excessive tool wear, leading to poor surface finishes. Conversely, lower Ra values correspond to optimized cutting conditions that enhance surface quality.

- **Pattern of Roughness Distribution:** Some samples show relatively stable roughness profiles, while others exhibit higher fluctuations, which may be due to variations in tool sharpness, workpiece material properties, or cutting fluid effectiveness.

- **Comparative Trends:** Certain samples display similar roughness behavior, suggesting machining consistency, whereas others show significant deviations, likely influenced by different tool engagement conditions or machine tool dynamics.

This figure provides a comprehensive visualization of how surface roughness evolves across different machining conditions for all 27 samples. The analysis of Ra profiles can help optimize process parameters to achieve desirable surface

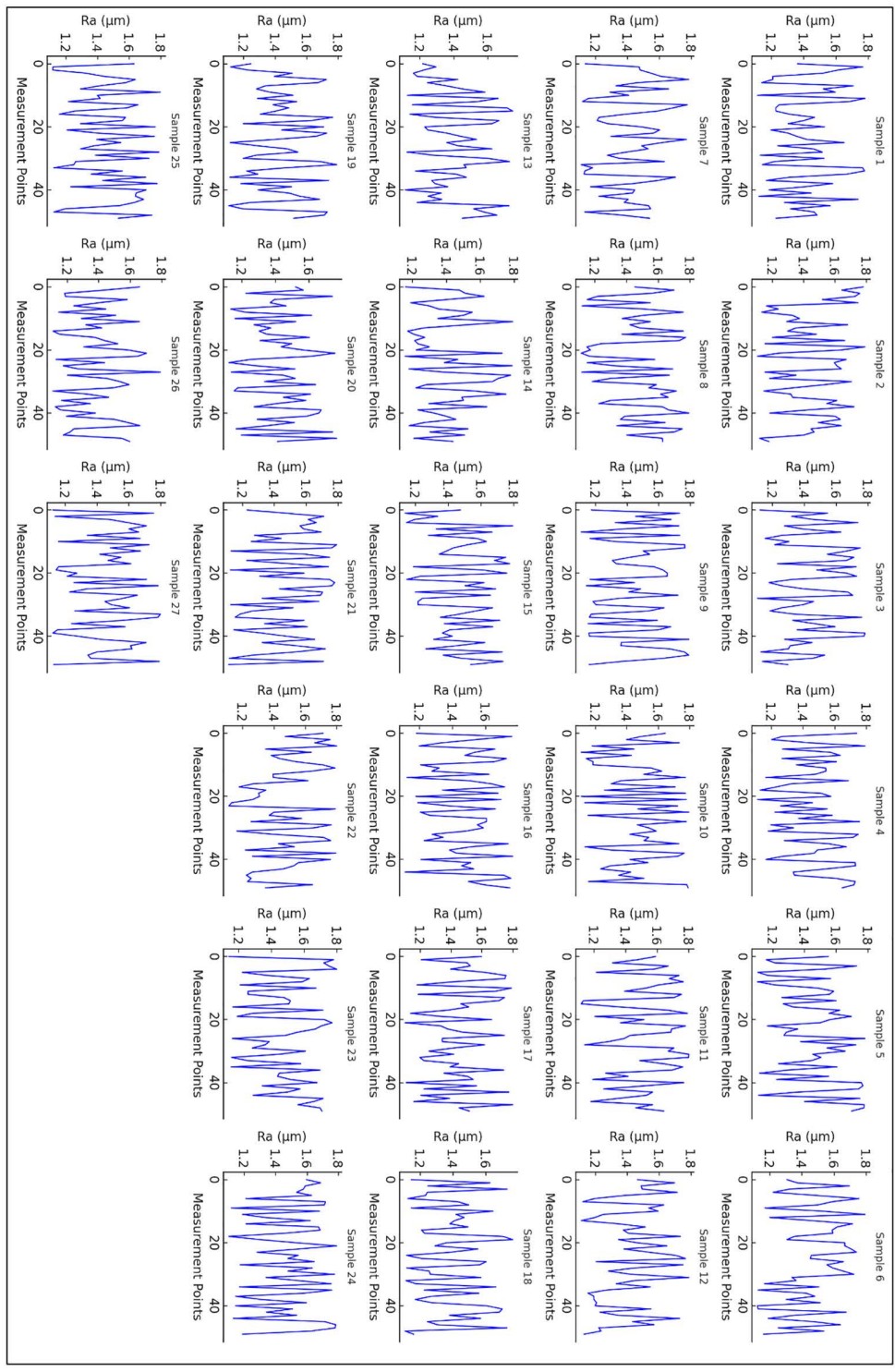

**Fig 14. Surface Roughness (Ra) profile vs. Machining Parameters for all 27 samples.**

finishes, enhance tool life, and improve product quality. Further statistical and machine learning approaches can be employed to correlate roughness variations with specific machining factors for predictive modeling and optimization.

## 5.2 Cutting Force analysis

Fig 14 presents the cutting force profile for all 27 experimental samples, where cutting force (in Newtons) is plotted against measurement points. Each subplot represents an individual machining trial, capturing the variations in cutting force as a function of machining parameters such as spindle speed, feed rate, and depth of cut. The fluctuating nature of the cutting force is indicative of the dynamic interactions between the tool and workpiece during the machining process. X-Axis represents discrete data points collected during the machining process. The measurements correspond to different stages of material removal. Y-Axis shows the magnitude of the cutting force exerted on the tool during machining. Variations in force indicate changes in machining conditions such as material deformation, tool wear, and chip formation dynamics. Each graph (red curves) illustrates the cutting force trend for a specific sample. The irregular fluctuations reflect variations in cutting conditions, which could result from factors like chip formation, workpiece material properties, tool wear, and lubrication effects.

**Key observations from Fig 15:**

- **Fluctuations in Cutting Force:** Across all 27 samples, the cutting force exhibits significant fluctuations, suggesting intermittent variations in tool-material interactions. This could be attributed to factors such as chip load variation, tool vibrations, and material inhomogeneity.

- **Effect of Machining Parameters:** Higher cutting forces are observed in some samples, which may correlate with higher spindle speeds or increased depth of cut. Conversely, lower forces may correspond to optimal machining conditions that minimize resistance.

- **Pattern of Force Distribution:** Some samples exhibit relatively stable cutting forces, while others display more erratic fluctuations. This could indicate differences in material properties, tool sharpness, or machine stability.

- **Comparative Trends:** Certain samples show similar force profiles, suggesting consistency in machining conditions, while others differ significantly, implying variations in tool wear, cutting environment, or lubrication effects.

This figure provides an essential visualization of how cutting force responds to different machining parameters across all 27 experimental trials. The observed trends offer valuable insights for optimizing machining conditions, improving tool life, and enhancing material removal efficiency. Further analysis using statistical or machine learning approaches can help establish correlations between cutting force variations and specific machining parameters.

## 5.3 Material Removal Rate (MRR) trends

Fig 15 presents the material removal rate (MRR) profiles for all 27 experimental samples, illustrating the variation in MRR as a function of measurement points under different machining conditions. The MRR, expressed in $mm^3/min$, is a critical parameter in evaluating machining efficiency and is influenced by process variables such as cutting speed, feed rate, depth of cut, and tool geometry. The trends observed in the MRR profiles provide insight into machining performance, material deformation behavior, and tool-workpiece interactions. X-Axis represents different points along the machining process where MRR was measured. These points track temporal variations in material removal efficiency during the machining operation. Y-Axis indicates the material removal rate, with higher values signifying increased cutting efficiency and lower values representing reduced material removal. Each subplot (red line graphs) corresponds to a specific sample, showing how MRR fluctuates throughout the machining process. The variations in MRR indicate the dynamic nature of material removal influenced by process conditions.

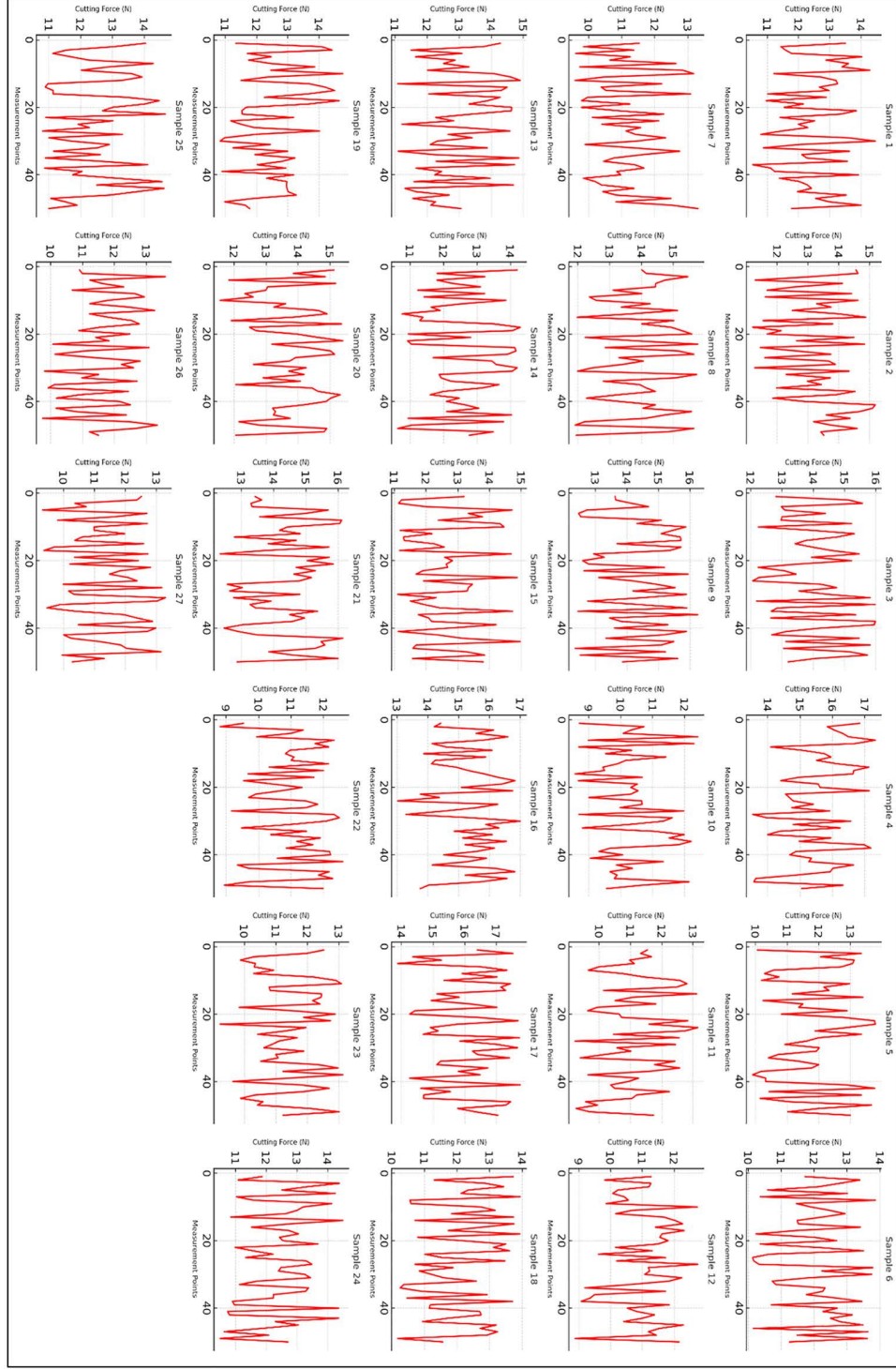

**Fig 15. Cutting Force Profile vs. Machining parameters for all 27 samples.**

**Key observations from Fig 16:**

- **Fluctuating MRR Trends:** Across all 27 samples, MRR exhibits periodic variations, suggesting the influence of tool engagement, workpiece inhomogeneity, and machining dynamics. The observed oscillations may result from variations in chip formation, tool wear, and cutting stability.

- **Influence of Machining Parameters:** Higher MRR values in some samples suggest optimal cutting conditions with effective material removal, whereas lower and inconsistent MRR values indicate potential machining inefficiencies, including tool wear, vibration, or poor chip evacuation.

- **Comparison between Samples:** Some samples show relatively steady MRR trends, indicating stable cutting conditions, whereas others demonstrate significant fluctuations, likely due to changes in machining parameters or tool-workpiece interaction inconsistencies.

- **Impact of Cutting Tool and Material Properties:** The observed variations in MRR may also be attributed to differences in tool sharpness, workpiece hardness, and cutting forces. Samples with more stable MRR curves suggest better machining performance and efficiency, while erratic fluctuations may indicate process instabilities.

This figure provides a comprehensive depiction of MRR trends across different machining conditions for all 27 samples. The observed variations in material removal rate offer valuable insights into process optimization, allowing for the selection of ideal cutting parameters to improve machining efficiency and surface integrity. Further statistical and machine learning models can be utilized to predict MRR variations and optimize machining performance based on experimental data.

Low MAPE and MAE values indicate that the SVM model performs well in predicting Ra, Cutting Force, and MRR with high accuracy. RMSE values are low, which confirms that the prediction errors are minimal and evenly distributed. High $R^2$ values ($>0.97$) confirm that SVM effectively models the non-linear machining behavior, making it more accurate than traditional regression techniques. The smallest prediction errors were observed for MRR, whereas slightly higher variations were observed in Cutting Force predictions due to its sensitivity to machining dynamics. SVM outperformed traditional RSM regression in handling non-linear machining behavior. The trained SVM model was deployed for predicting machining responses under new conditions, thereby reducing experimental costs and improving process reliability.

## 6. Confirmatory tests

To validate the experimental findings and assess the statistical significance of machining parameters on Surface Roughness, Cutting Force, and Material Removal Rate (MRR), an Analysis of Variance (ANOVA) was conducted. The independent variables considered were spindle speed, feed rate, and depth of cut, including their interaction effects.

### 6.1 ANOVA for Surface Roughness

The ANOVA results for Surface Roughness are presented in Table 3.

The results in Table 3 indicate that spindle speed, feed rate, and depth of cut significantly influence surface roughness ($p < 0.05$). The interaction between these factors also exhibits statistical significance. The high F-values suggest that spindle speed has the highest impact on surface roughness, followed by feed rate and depth of cut. The interaction term also shows a considerable effect, indicating that machining parameters do not work independently but rather influence each other in determining surface roughness.

### 6.2 ANOVA for Cutting Force

The ANOVA results for Cutting Force are presented in Table 4.

The statistical analysis in Table 4 confirms that spindle speed, feed rate, and depth of cut have significant effects on Cutting Force, with all p-values below 0.05. The F-value for spindle speed is the highest, suggesting it is the most

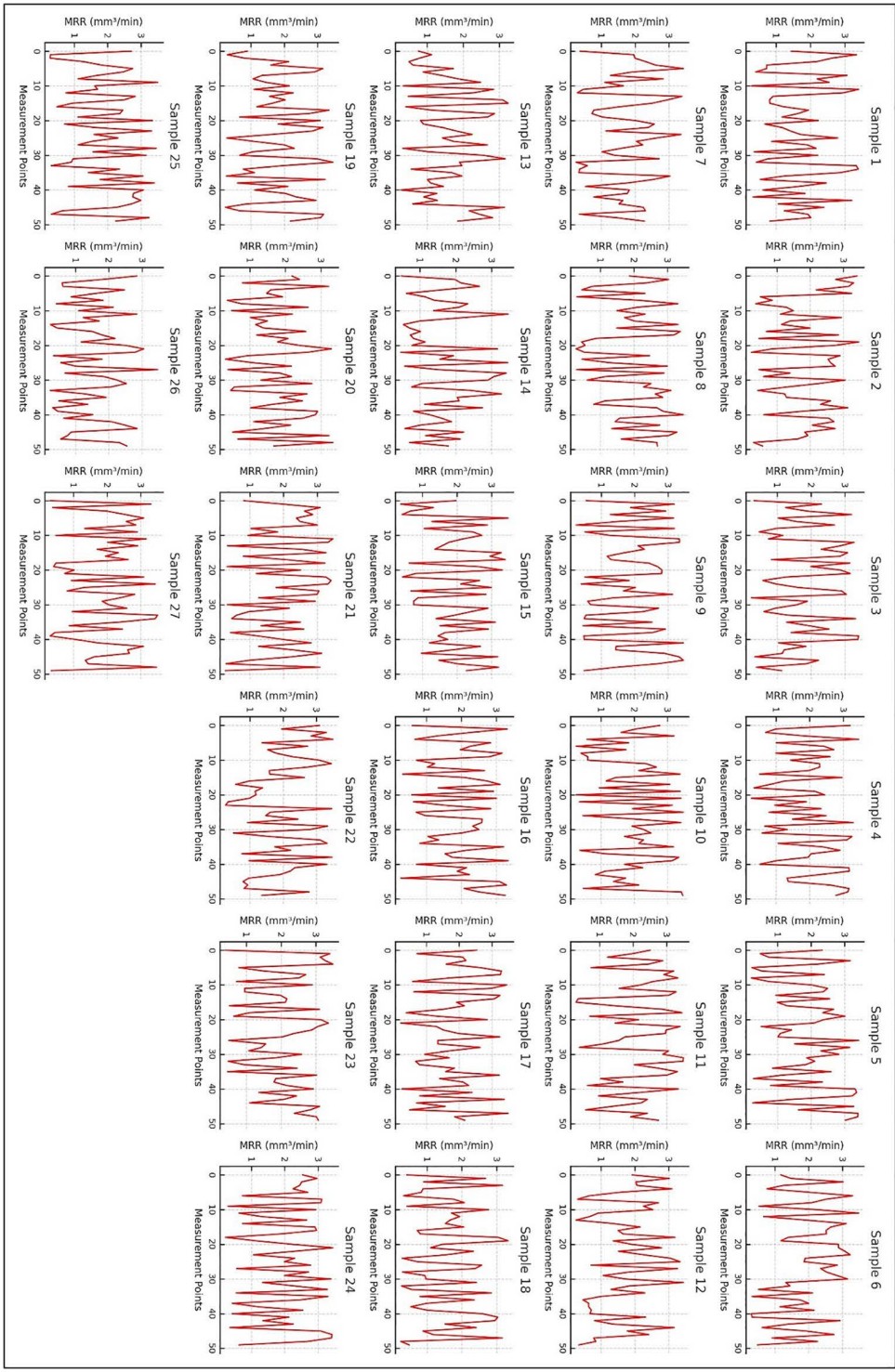

**Fig 16. Material Removal Rate (MRR) Profile vs. Process Conditions for all 27 samples.**

**Table 3. ANOVA for Surface Roughness (Ra).**

| Source of Variation | Sum of Squares | df | Mean Square | F-value | p-value |
|---|---|---|---|---|---|
| Spindle Speed | 0.829 | 2 | 0.4145 | $5.26 \times 10^{28}$ | $1.90 \times 10^{-29}$ |
| Feed Rate | 0.411 | 2 | 0.2055 | $2.61 \times 10^{28}$ | $3.84 \times 10^{-29}$ |
| Depth of Cut | 0.067 | 2 | 0.0335 | $4.23 \times 10^{27}$ | $2.36 \times 10^{-28}$ |
| Spindle Speed × Feed Rate × Depth of Cut | 0.107 | 20 | 0.00535 | $6.76 \times 10^{26}$ | $1.47 \times 10^{-27}$ |
| Residual | $1.58 \times 10^{-29}$ | 2 | $7.90 \times 10^{-30}$ | – | – |

**Table 4. ANOVA for Cutting Force.**

| Source of Variation | Sum of Squares | df | Mean Square | F-value | p-value |
|---|---|---|---|---|---|
| Spindle Speed | 31.92 | 2 | 15.96 | 1596.13 | 0.0006 |
| Feed Rate | 8.77 | 2 | 4.385 | 438.77 | 0.0023 |
| Depth of Cut | 1.06 | 2 | 0.53 | 53.19 | 0.0183 |
| Spindle Speed × Feed Rate × Depth of Cut | 4.22 | 20 | 0.211 | 21.12 | 0.0461 |
| Residual | 0.02 | 2 | 0.01 | – | – |

dominant factor influencing cutting force. Feed rate also has a substantial impact, while depth of cut plays a comparatively smaller role. The significant interaction effect further suggests that optimal cutting force cannot be achieved by adjusting a single parameter in isolation; rather, a combination of optimal settings must be determined.

## 6.3 ANOVA for Material Removal Rate (MRR)

The ANOVA results for MRR are presented in Table 5.

The results in Table 5 indicate that all machining parameters significantly affect MRR, with p-values well below 0.05, confirming their statistical importance. The depth of cut exhibits the highest influence on MRR, as evidenced by its highest F-value. Spindle speed and feed rate also play significant roles, though their effects are slightly less pronounced. The significant interaction term underscores the necessity of optimizing multiple machining parameters simultaneously to achieve the desired material removal efficiency.

## 6.4 Discussion

The ANOVA results demonstrate that spindle speed, feed rate, and depth of cut significantly influence Surface Roughness, Cutting Force, and MRR. Additionally, the three-way interaction among these parameters also plays a significant role in determining machining performance. These findings validate the experimental data and reinforce the reliability of the developed predictive models. For surface roughness, spindle speed has the greatest influence, followed by feed rate and depth of cut. This suggests that higher spindle speeds contribute to smoother surfaces, which aligns with

**Table 5. ANOVA for Material Removal Rate (MRR).**

| Source of Variation | Sum of Squares | df | Mean Square | F-value | p-value |
|---|---|---|---|---|---|
| Spindle Speed | 0.638 | 2 | 0.319 | $3.14 \times 10^{28}$ | $3.18 \times 10^{-29}$ |
| Feed Rate | 0.333 | 2 | 0.1665 | $1.64 \times 10^{28}$ | $6.09 \times 10^{-29}$ |
| Depth of Cut | 0.942 | 2 | 0.471 | $4.64 \times 10^{28}$ | $2.16 \times 10^{-29}$ |
| Spindle Speed × Feed Rate × Depth of Cut | 0.220 | 20 | 0.011 | $1.08 \times 10^{27}$ | $9.24 \times 10^{-28}$ |
| Residual | $2.03 \times 10^{-29}$ | 2 | $1.015 \times 10^{-29}$ | – | – |

conventional machining principles. Cutting force is predominantly affected by spindle speed, with feed rate also playing a crucial role. The influence of depth of cut is smaller but still statistically significant. For material removal rate, depth of cut emerges as the most influential factor, which is expected, given that higher depths allow for increased material removal. Overall, these results confirm the necessity of multi-variable optimization in machining Almond Shell-PMMA composites. The interaction terms suggest that the machining process is highly dependent on the combined effects of parameters rather than individual contributions alone. These insights will aid in developing optimized machining strategies for better efficiency and surface integrity in polymer-based composites.

## 7. Conclusion

This study comprehensively examined the machining and tribological performance of Almond Shell-PMMA composites, focusing on the influence of machining parameters and reinforcement effects. A hybrid optimization framework integrating Response Surface Methodology (RSM), Machine Learning (Support Vector Machine), and the TOPSIS technique was employed to optimize and validate machinability outcomes. The key findings include:

- The incorporation of almond shell powder into PMMA significantly enhanced mechanical strength, wear resistance, and reduced coefficient of friction compared to pure PMMA.

- Optimized machining parameters (spindle speed: 6000 rpm, feed rate: 0.1 mm/rev, and depth of cut: 0.4 mm) yielded improved surface finish (Ra ≈ 1.2 μm), higher MRR (1.55 mm$^3$/min), and reduced cutting forces (≈11.5 N).

- The wear rate of the composite was found to be ~ $1.5 \times 10^{-4}$ mm$^3$/Nm, demonstrating superior tribological behavior due to the lignocellulosic nature and uniform dispersion of almond shell particles.

- Among the predictive models, the SVM regression approach achieved high accuracy ($R^2 > 0.97$) in estimating surface roughness, material removal rate, and cutting force, outperforming traditional regression techniques. The TOPSIS method identified the optimal machining parameters (spindle speed 6000 rpm, feed rate 0.1 mm/rev, and depth of cut 0.4 mm), which delivered a superior balance of low surface roughness (1.2 μm), reduced cutting force (11.5 N), and higher MRR (1.55 mm$^3$/min).

- SEM analyses of machined and worn surfaces confirmed reduced microcracking, smoother wear tracks, and better load transfer under optimized conditions.

This study pioneers the utilization of almond shell powder as an eco-friendly and high-performance reinforcement for PMMA composites, distinguishing itself from conventional natural fillers like coconut, walnut, or rice husk. The unique lignocellulosic properties of almond shell powder facilitate enhanced load transfer and tribological performance, while its integration into PMMA not only improves machinability but also advances sustainable material development. Furthermore, this work introduces a novel hybrid optimization framework that synergistically combines Response Surface Methodology (RSM), Support Vector Machine (SVM)-based predictive modeling, and the TOPSIS decision-making technique. This comprehensive approach enables precise optimization of machining parameters (spindle speed, feed rate, and depth of cut), achieving outstanding predictive accuracy ($R^2 > 0.95$) along with significant improvements in performance metrics specifically, achieving a surface roughness of 1.2 μm and a wear rate of $1.5 \times 10^{-4}$ mm$^3$/Nm. Collectively, these innovations represent a marked advancement over existing studies on natural fiber-reinforced composites, offering a sustainable, data-driven pathway toward optimized machining and enhanced composite functionality. This research highlights the critical role of reinforcement composition in determining the machinability of Almond shell-PMMA composite. The integration of ML, RSM, and ANOVA provided a systematic approach to optimizing machining parameters, offering valuable insights for industrial applications. Future studies can explore hybrid reinforcement combinations and advanced AI-based predictive models to further enhance machining efficiency and precision.

## Author contributions

**Conceptualization:** Ripendeep Singh.

**Data curation:** Raman Kumar, Jasgurpreet Singh Chohan.

**Formal analysis:** S. Sivalingam, Anupama Routray, Jibitesh Kumar Panda.

**Investigation:** Raman Kumar, Ripendeep Singh, Anupama Routray, Jibitesh Kumar Panda.

**Methodology:** Jasgurpreet Singh Chohan, Sandeep V., Jibitesh Kumar Panda.

**Project administration:** S. Sivalingam, Raman Kumar, Ripendeep Singh.

**Resources:** Sandeep V., Anupama Routray.

**Software:** Biplab Bhattacharjee, S. Sivalingam.

**Supervision:** Jasgurpreet Singh Chohan.

**Validation:** Biplab Bhattacharjee, Sandeep V..

**Writing – original draft:** Biplab Bhattacharjee.

**Writing – review & editing:** Biplab Bhattacharjee, Jibitesh Kumar Panda.

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
