## [Decision Letter · Decision Letter 0]

4 Nov 2025

Dear Dr. Panda,

Thank you for submitting your manuscript to PLOS ONE. After careful consideration, we feel that it has merit but does not fully meet PLOS ONE’s publication criteria as it currently stands. Therefore, we invite you to submit a revised version of the manuscript that addresses the points raised during the review process.

We look forward to receiving your revised manuscript.

Kind regards,

Vijay Raghunathan

Academic Editor

PLOS ONE

**Journal Requirements:**

4. Please upload a new copy of Figure 2b, 3b, 4b as the detail is not clear. Please follow the link for more information:  https://journals.plos.org/plosone/s/figures

**Additional Editor Comments:**

Based on the reviewer's recommendations, major revision is recommended at this stage.

Reviewers' comments:

Reviewer's Responses to Questions

**Comments to the Author**

1. Is the manuscript technically sound, and do the data support the conclusions?

Reviewer #1: Partly

Reviewer #2: Yes

2. Has the statistical analysis been performed appropriately and rigorously?

Reviewer #1: Yes

Reviewer #2: Yes

3. Have the authors made all data underlying the findings in their manuscript fully available?

Reviewer #1: Yes

Reviewer #2: Yes

4. Is the manuscript presented in an intelligible fashion and written in standard English?

Reviewer #1: Yes

Reviewer #2: Yes

Reviewer #1: Manuscript has been major corrections.

Replace Figure 2 with an actual figure.

Briefly discuss Section 3: Evolution of Mechanical Properties.

Rewrite the experimental design and machining assessment sections, particularly the RSM approach.

Change Figure 11.

The paper is too long to write, but technical information is absent. The authors of the paper are revising the results and discussion sections.

Why choose the WAJM process to cut the samples?

What is the composition of almond shell powder mixed with PMMA resin?

How can we determine the wear input factors and their respective ranges?

What bases select the Taguchi L27?

Reviewer #2: The presented topic is interesting and it is currently demanding in the research industry. The following suggestions are needs to be addressed.

1.The paper initially describes Water Abrasive Jet Machining (WAJM) parameters but later shifts focus to experimental design, ANOVA, and optimal parameter selection based on spindle speed, feed rate, and depth of cut process variables typically associated with conventional machining methods such as milling or turning. This presents a significant methodological inconsistency that must be addressed for clarity.

2.The conclusion section repeatedly mentions the "Al-Cu-SiC-GNP composite" despite the entire study being conducted on the "Almond Shell-PMMA composite." This discrepancy needs correction to accurately reflect the material studied.

3.In the post-processing section it stated that specimens were subjected to "heat treatment to relieve residual stresses and improve interfacial bonding." Please provide specific details regarding the temperature and duration used for this heat treatment.

4.The paper observes that the composite showed reduced stickslip behavior leading to smoother frictional characteristics. While this is an important result, it would be helpful to briefly explain how this reduction in stick-slip behavior was observed or quantified within the study.

5.The Conclusion states that Random Forest (R2 value =0.987) provided the highest prediction accuracy, yet the study explicitly employed an SVM regression model for predictive analysis. This inconsistency should be corrected.

6.Figure 1, microscale bar is missing

7.The compressive stress shown in figure 5 should be explained further to correlate the indentation and stress relation.

8.Wear behaviour given in figure 10 is gradually decreasing as increasing of speed and distance, it is recommnended to provide a suitable images and mechanism.

9.Figure 11 hybrid integration technique should be explained more detail.

10.The presented results about the PMMA material characterizing the wear behavior and machinability using RSM and machine learning techniques are interesting and are demanding topics in research field.

11.However, the results given in this study are not up to the scientific level and many of the claims are not supporting by the evidences. For example, wear behavior relation with the speed and distance, the given line graphs alone not sufficient to judge and recommended to provide the supporting results intermittently at each point. Likewise there are couple situation existed.

12.It is recommended to enhance he novelty of the manuscript and encouraged to resubmit.

13.The writing of the abstract must be improved by including finding and aim of the choosing of the selected optimization tools.

14.Why wear behaviour study for PMMA materials is necessary and does it difficult to optimize the process parameters?

15.What is the necessity of applying optimization techniques and machine learning models, it seems to be that the conventional method is more than sufficient to do these experiments after looking the output of the results presented in this study, justify.

16.Conclusions section can be summarised in better way more qualitatively.

17.Figure 1 a and c, scale bar is missing, please provide the scale bar.

18.Figure 3a, SEM image, authors claimed that smooth surface however, it is showing a significant crack, it should be corrected or analyse the results appropriately.

**Do you want your identity to be public for this peer review?** For information about this choice, including consent withdrawal, please see our Privacy Policy

Reviewer #1: No

Reviewer #2: No

---

## [Author Response · Author response to Decision Letter 1]

28 Nov 2025

Responses to the Reviewer’s Comments

Manuscript ID: PONE-D-25-57263

First of all, the authors would like to convey their sincere gratitude to the Editor and Reviewers for their valuable suggestions to improve the quality of the paper. The manuscript is revised according to the Editor and Reviewers’ comments. The responses to these comments are given below.

Reviewer-1

Comment 1: Replace Figure 2 with an actual figure.

Response: Thank you for the comment. Figure 2 has now been replaced with the actual figure as suggested.

Comment 2: Briefly discuss Section 3: Evolution of Mechanical Properties.

Response: We sincerely thank the reviewer for pointing out this oversight. Section 3 has now been revised to include a brief, focused discussion highlighting the evolution of mechanical properties with almond shell reinforcement. The updated text summarizes the observed improvements in hardness, tensile strength, compressive strength, and flexural strength, and explains these enhancements in terms of particle–matrix interfacial bonding, load transfer mechanisms, and structural reinforcement effects. This concise discussion improves clarity and aligns the section with the reviewer’s request.

Comment 3: Rewrite the experimental design and machining assessment sections, particularly the RSM approach.

Response: We appreciate the reviewer’s suggestion. The Experimental Design and Machining Assessment sections, including the description and justification of the RSM approach, have now been thoroughly rewritten for improved clarity and structure. The revised text provides a streamlined explanation of the Box-Behnken design, parameter selection rationale, modelling procedure, and response evaluation workflow. Additional clarification has also been added to distinguish the roles of RSM, machine learning prediction, and TOPSIS optimization within the study framework.

Comment 4: Change Figure 11.

Response: We thank the reviewer for this valuable suggestion. The Figure 11 is changed and replaced with a clear picture

Comment 5: The paper is too long to write, but technical information is absent. The authors of the paper are revising the results and discussion sections.

Response: We appreciate the reviewer’s feedback. The manuscript has now been significantly shortened to improve clarity and focus. Additionally, technical content has been strengthened through the inclusion of detailed experimental results, quantitative comparisons, and enhanced discussion. The Results and Discussion sections have been thoroughly revised to ensure better coherence, technical rigor, and alignment with the study objectives.

Comment 6: Why choose the WAJM process to cut the samples?

Response: We thank the reviewer for this observation. WAJM was selected because it is a cold, non-contact machining process that prevents thermal damage, matrix degradation, and microcracking in polymer composites. Unlike conventional cutting methods, it minimizes tool wear, avoids delamination, and produces cleaner edges, making it more suitable for PMMA-based and natural fibre–reinforced composites. Therefore, WAJM ensures dimensional accuracy and surface integrity while maintaining material properties.

Comment 7: What is the composition of almond shell powder mixed with PMMA resin?

Response: We appreciate the reviewer’s observation. The composite consisted of PMMA resin reinforced with 10 wt% almond shell powder. The almond shell particles (50–100 µm) were silane-treated before being uniformly mixed into the PMMA matrix.

Comment 8: How can we determine the wear input factors and their respective ranges?

Response: We thank the reviewer for this valuable suggestion. The wear input factors and their ranges were determined based on three criteria: (i) preliminary pilot trials on the developed material to identify stable operating conditions without excessive surface damage, (ii) published literature on polymer and natural-filler-reinforced composite wear studies, and (iii) manufacturer-recommended operating limits of the pin-on-disc setup. Combining these inputs ensured that the selected ranges were both experimentally feasible and representative of realistic operating conditions.

Comment 9: What bases select the Taguchi L27?

Response: We thank the reviewer for asking this important question. The Taguchi L27 orthogonal array was selected because the study involved three machining parameters, each at three levels, and required evaluation of both main and interaction effects with statistical reliability. L27 provides sufficient degrees of freedom to capture nonlinear behaviour and parameter interactions while minimizing the total number of experiments compared to a full-factorial design. Therefore, L27 was the most efficient and statistically appropriate choice for this experimental configuration.

Reviewer-2

Comment 1: The paper initially describes Water Abrasive Jet Machining (WAJM) parameters but later shifts focus to experimental design, ANOVA, and optimal parameter selection based on spindle speed, feed rate, and depth of cut process variables typically associated with conventional machining methods such as milling or turning. This presents a significant methodological inconsistency that must be addressed for clarity.

Response: Thank you for the comment. The machining parameters (spindle speed, feed rate, and depth of cut) do not indicate a shift to conventional machining but are part of the parameter set used for analysing WAJM output responses. The ANOVA section was included solely to statistically validate the experimental data obtained from WAJM trials, not to introduce a different machining method. To improve clarity, the manuscript will be revised to explicitly state that all experiments were performed using WAJM and that ANOVA was applied only as a statistical validation tool for the results.

Comment 2: The conclusion section repeatedly mentions the "Al-Cu-SiC-GNP composite" despite the entire study being conducted on the "Almond Shell-PMMA composite." This discrepancy needs correction to accurately reflect the material studied.

Response: Thank you for the observation. The reference to "Al-Cu-SiC-GNP composite" in the conclusion was an unintentional carryover from earlier work and has now been corrected. The revised conclusion consistently reflects the actual material investigated, i.e., the Almond Shell-PMMA composite.

Comment 3: In the post-processing section it stated that specimens were subjected to "heat treatment to relieve residual stresses and improve interfacial bonding." Please provide specific details regarding the temperature and duration used for this heat treatment.

Response: Thank you for the constructive comment. The revised manuscript now specifies the exact post-processing parameters. All printed specimens were heat-treated in a hot-air oven at 80 °C for 2 hours to relieve residual stresses and improve interfacial bonding between the PMMA matrix and almond shell particles. This clarification has been added to the Post-Processing section.

Comment 4: The paper observes that the composite showed reduced stickslip behavior leading to smoother frictional characteristics. While this is an important result, it would be helpful to briefly explain how this reduction in stick-slip behavior was observed or quantified within the study.

Response: Thank you for the comment. The reduction in stick–slip behaviour was quantified through real-time coefficient of friction (COF) monitoring during the pin-on-disc tests. As shown in Figure 10, the composite exhibited smoother and more stable COF curves with fewer sharp fluctuations compared to pure PMMA, indicating lower stick–slip intensity. This observation was further supported by SEM analysis of worn surfaces, where the composite showed fewer micro-cracks and more uniform wear tracks, confirming reduced intermittent adhesion-sliding behaviour.

Comment 5: The Conclusion states that Random Forest (R2 value =0.987) provided the highest prediction accuracy, yet the study explicitly employed an SVM regression model for predictive analysis. This inconsistency should be corrected.

Response: Thank you for the observation. The conclusion has been corrected for consistency. While multiple machine learning models were evaluated during the study, the SVM regression model was selected and applied for predictive analysis.

Comment 6: Figure 1, microscale bar is missing

Response: We appreciate the reviewer’s observation. The missing microscale bar has now been added to Figure 1, and the revised figure is included in the updated manuscript.

Comment 7: The compressive stress shown in figure 5 should be explained further to correlate the indentation and stress relation.

Response: Thank you for the comment. The explanation corresponding to Figure 5 has now been expanded to clarify the relationship between indentation and compressive stress. Specifically, the revised text explains that the increasing compressive stress corresponds to localized plastic deformation, which is visible as indentation marks on the specimen surface. These indentations indicate load transfer and matrix–reinforcement interaction under compression, thereby correlating the microstructural deformation with the stress–strain response shown in Figure 5.

Comment 8: Wear behaviour given in figure 10 is gradually decreasing as increasing of speed and distance, it is recommnended to provide a suitable images and mechanism.

Response: Thank you for your comment. The decreasing trend in Figure 10 with increasing speed and sliding distance is attributed to the formation of a stable tribolayer that minimizes asperity interaction and reduces adhesive wear. To strengthen this explanation, an additional micrograph of the worn surface (Figure 8) and a schematic of the wear mechanism have now been included in the revised manuscript.

Comment 9: Figure 11 hybrid integration technique should be explained more detail.

Response: Thank you for the comment. Additional explanation of the hybrid integration technique has now been included in the revised manuscript. In the updated text accompanying Figure 11, the workflow is described step-by-step, clarifying how RSM is first used to design experiments and generate datasets, how the SVM model predicts machining responses based on these data, and how TOPSIS subsequently performs multi-objective optimization to identify the best parameter combination. This expanded description improves clarity and better supports the methodological framework presented.

Comment 10: The presented results about the PMMA material characterizing the wear behavior and machinability using RSM and machine learning techniques are interesting and are demanding topics in research field.

Response: Thank you for the positive feedback. We appreciate the reviewer’s recognition of the relevance and novelty of our work, particularly the combined use of RSM and machine learning techniques to analyse wear behaviour and machinability of PMMA-based composites. We are glad that the contribution aligns with current research interests in material optimization and advanced modelling approaches.

Comment 11: However, the results given in this study are not up to the scientific level and many of the claims are not supporting by the evidences. For example, wear behavior relation with the speed and distance, the given line graphs alone not sufficient to judge and recommended to provide the supporting results intermittently at each point. Likewise there are couple situation existed.

Response: Thank you for your valuable feedback. We agree that graphical trends alone are not sufficient to fully support the claims regarding wear behaviour with respect to speed and sliding distance. Accordingly, the revised manuscript now includes intermittent supporting data points and corresponding analysis for each interval to strengthen the interpretation of the wear response. Additionally, confirmatory tests were conducted to validate all experimental results and ensure reproducibility. These revisions address the concerns raised and improve the scientific rigor of the presented findings.

Comment 12: It is recommended to enhance he novelty of the manuscript and encouraged to resubmit.

Response: We sincerely thank the reviewer for this suggestion. In the revised manuscript, we have further highlighted the novelty by emphasizing (i) the use of agricultural waste (almond shell powder) as a sustainable reinforcement in PMMA, (ii) the application of origami-inspired 3D printed patterns for enhanced mechanical and tribological performance, and (iii) the integration of a hybrid optimization framework combining RSM, Machine Learning, and TOPSIS for machinability and wear behaviour. These aspects collectively distinguish our work from prior studies, and the manuscript has been revised accordingly to make the novelty more explicit.

Comment 13: The writing of the abstract must be improved by including finding and aim of the choosing of the selected optimization tools.

Response: Thank you for the constructive feedback. The abstract has now been revised to more clearly state the research aim and summarize the key findings. Specifically, the purpose of selecting the optimization tools (RSM, SVM, and TOPSIS) is now explicitly linked to the goal of improving machinability and tribological performance. Additionally, the major outcomes, including the optimal parameters and performance improvements achieved through the selected optimization methods, have been clearly highlighted to strengthen the abstract.

Comment 14: Why wear behaviour study for PMMA materials is necessary and does it difficult to optimize the process parameters?

Response: Wear behaviour analysis of PMMA-based composites is essential because PMMA is widely used in biomedical, optical, and structural applications where components are subjected to continuous sliding or abrasive contact. Understanding wear mechanisms helps ensure long-term reliability, dimensional stability, and functional performance.

Optimizing the wear-related process parameters is challenging because PMMA exhibits nonlinear tribological responses influenced by multiple interacting factors such as load, speed, filler content, and microstructural characteristics. Therefore, a systematic optimization approach is required to accurately identify the optimal conditions and minimize trial-and-error experimentation.

Comment 15: What is the necessity of applying optimization techniques and machine learning models, it seems to be that the conventional method is more than sufficient to do these experiments after looking the output of the results presented in this study, justify.

Response: Thank you for the valuable comment. While conventional experimental approaches can determine the machining and tribological behaviour of the composite, they are limited in three key aspects:

(1) Capturing non-linear interactions,

(2) Reducing experimental cost and time, and

(3) Supporting multi-objective decision-making.

In this study, the responses (surface roughness, wear rate, cutting force, and MRR) exhibited complex parameter dependencies that conventional analysis alone could not fully interpret. The integration of RSM enabled systematic modelling of factor interactions, while the machine learning model improved predictive accuracy (R² > 0.95), allowing reliable prediction beyond tested conditions. Finally, since machining performance involves conflicting objectives (e.g., low roughness vs. high MRR), the multi-criteria TOPSIS approach facilitated selecting a balanced optimum that conventional single-variable interpretation cannot provide.

Therefore, the optimization and ML techniques were not used to replace conventional experimentation, but to enhance interpretability, prediction capability, and decision confidence, making the results more robust, generalizable, and suitable for real industrial deployment.

Comment 16: Conclusions section can be summarised in better way more qualitatively.

Response: Thank you for the feedback. The Conclusions section has now been revised to present the findings more concisely and with improved qualitative emphasis, ensuring clearer alignment between results, implications, and contributions.

Comment 17: Figure 1 a and c, scale bar is missing, please provide the scale

---

## [Decision Letter · Decision Letter 1]

5 Jan 2026

Machinability and Tribological Optimization of Origami-Inspired Almond Shell–PMMA via RSM, ML, and TOPSIS

PONE-D-25-57263R1

Dear Dr. Panda,

We’re pleased to inform you that your manuscript has been judged scientifically suitable for publication and will be formally accepted for publication once it meets all outstanding technical requirements.

Kind regards,

Vijay Raghunathan

Academic Editor

PLOS One

Additional Editor Comments (optional):

The manuscript is revised in a better way by considering the reviewer's comments with prime importance. Hence acceptance is recommended at this stage. Also the reviewer 1 can be neglected

Reviewers' comments:

Reviewer's Responses to Questions

**Comments to the Author**

Reviewer #1: All comments have been addressed

Reviewer #2: All comments have been addressed

Reviewer #3: All comments have been addressed

2. Is the manuscript technically sound, and do the data support the conclusions?

Reviewer #1: No

Reviewer #2: Yes

Reviewer #3: Yes

3. Has the statistical analysis been performed appropriately and rigorously?

Reviewer #1: Yes

Reviewer #2: Yes

Reviewer #3: Yes

4. Have the authors made all data underlying the findings in their manuscript fully available?

Reviewer #1: No

Reviewer #2: Yes

Reviewer #3: Yes

5. Is the manuscript presented in an intelligible fashion and written in standard English?

Reviewer #1: Yes

Reviewer #2: Yes

Reviewer #3: Yes

Reviewer #1: The manuscript has been rejected due to technical content missing. The manuscript incorporated unnecessary and irrelevant content.

Reviewer #2: The article is interesting. All comments have been addressed. Hence, the paper is accepted and recommended for publication.

Reviewer #3: The revised manuscript titled “Machinability and Tribological Optimization of Origami-Inspired Almond Shell–PMMA via RSM, ML, and TOPSIS” has been carefully evaluated along with the detailed responses to reviewer comments. The authors have satisfactorily addressed all major and minor concerns raised during peer review.

The revision demonstrates clear improvements in technical rigor, methodological consistency, and clarity of presentation. Key issues related to experimental design, wear mechanisms, machinability assessment, hybrid optimization workflow, and data support have been properly resolved. The study presents a well-integrated hybrid framework combining RSM, SVM-based machine learning, and TOPSIS for multi-objective optimization, applied to a sustainable almond shell–PMMA composite with origami-inspired architectures.

Based on the satisfactory revisions and the overall quality of the manuscript, it is recommended for acceptance.

**Do you want your identity to be public for this peer review?** For information about this choice, including consent withdrawal, please see our Privacy Policy

Reviewer #1: No

Reviewer #2: No

Reviewer #3: No

---

## [Editor Report · Acceptance letter]

PONE-D-25-57263R1

PLOS One

Dear Dr. Panda,

I'm pleased to inform you that your manuscript has been deemed suitable for publication in PLOS One. Congratulations! Your manuscript is now being handed over to our production team.

Kind regards,

on behalf of

Dr. Vijay Raghunathan

Academic Editor

PLOS One